# Quantum logic with spin qubits crossing the surface code threshold

Xiao Xue[1,2], Maximilian Russ[1,2], Nodar Samkharadze[1,3], Brennan Undseth[1,2], Amir Sammak[1,3], Giordano Scappucci[1,2] & Lieven M. K. Vandersypen[1,2 ✉]

High-fidelity control of quantum bits is paramount for the reliable execution of quantum algorithms and for achieving fault tolerance—the ability to correct errors faster than they occur[1]. The central requirement for fault tolerance is expressed in terms of an error threshold. Whereas the actual threshold depends on many details, a common target is the approximately 1% error threshold of the well-known surface code[2,3]. Reaching two-qubit gate fidelities above 99% has been a long-standing major goal for semiconductor spin qubits. These qubits are promising for scaling, as they can leverage advanced semiconductor technology[4]. Here we report a spin-based quantum processor in silicon with single-qubit and two-qubit gate fidelities, all of which are above 99.5%, extracted from gate-set tomography. The average single-qubit gate fidelities remain above 99% when including crosstalk and idling errors on the neighbouring qubit. Using this high-fidelity gate set, we execute the demanding task of calculating molecular ground-state energies using a variational quantum eigensolver algorithm[5]. Having surpassed the 99% barrier for the two-qubit gate fidelity, semiconductor qubits are well positioned on the path to fault tolerance and to possible applications in the era of noisy intermediate-scale quantum devices.

Quantum computation involves the execution of a large number of elementary operations that take a qubit register through the steps of a quantum algorithm[6]. A major challenge is to implement these operations with sufficient accuracy to arrive at a reliable outcome, even in the presence of decoherence and other error sources. The higher the accuracy, or fidelity, of the operations, the higher the likelihood that near-term applications for quantum computers come within reach[7]. Furthermore, for most presently known algorithms, the number of operations that must be concatenated will unavoidably lead to excessive accumulation of errors, and these errors must be removed using quantum error correction[1]. Correcting quantum errors faster than they occur is possible when the error probability per operation is below a certain threshold, known as the fault-tolerance threshold. For the widely considered surface code, for instance, the fault-tolerance threshold is between 0.6% and 1%, under certain assumptions, albeit at the cost of a large redundancy in the number of physical qubits[2,3].

Among all the candidate platforms, electron spins in semiconductor quantum dots have advantages, such as their long coherence times[8], small footprint[9], the potential for scaling up[10] and the compatibility with advanced semiconductor manufacturing technology[4]. Single-qubit operations of spin qubits in quantum dots achieve fidelities of 99.9% (refs. [11,12]) but the two-qubit gate fidelities reported vary from 92% to 98% (refs. [13,14]). This has limited the two-qubit Bell-state fidelities to 94% (ref. [15]) and quantum algorithms implemented with spin qubits gave only coarsely accurate outcomes[16,17]. Pushing the two-qubit gate fidelity well beyond 99% requires not only low charge-noise levels and the elimination of nuclear spins by isotopic enrichment but also careful Hamiltonian engineering.

In this paper, using a precisely engineered two-qubit interaction Hamiltonian, we report the demonstration of single-qubit and two-qubit gates with fidelities above 99.5%. We use gate-set tomography (GST) not only to characterize the gates and to quantify the fidelity but also to improve the gate calibration. The high-fidelity gates allow us to compute the dissociation energy of molecular hydrogen with a variational quantum eigensolver (VQE) algorithm, reaching an accuracy for the dissociation energy of around 20 mHa, limited by readout errors.

We use a gate-defined double quantum dot in an isotopically enriched [28]Si/SiGe heterostructure[17] (Fig. 1a), with each dot occupied by a single electron (see Methods). The spin states of the electrons serve as qubits. The spin states are measured with the help of a sensing quantum dot (SQD), which is capacitively coupled to the qubit dots[18]. A micromagnet on top of the device provides a magnetic field gradient enabling electric-dipole spin resonance[19] and separates the resonance frequencies of the qubits in the presence of an external magnetic field (~320 mT) to 11.993 GHz ($Q_1$) and 11.890 GHz ($Q_2$). Single-qubit X and Y gates are implemented by frequency-multiplexed microwave signals applied to gate MW and virtual Z gates are implemented by a phase update of the reference frame[20]. The plunger gates (LP and RP) control the chemical potentials of the quantum dots.

The native two-qubit gate for spin qubits uses the exchange interaction[21,22], originating from the wave-function overlap of electrons in neighbouring dots. This selectively shifts the energy of the antiparallel spin states and, thus, enables an electrically pulsed adiabatic conditional Z (CZ) gate[8,16,23]. The barrier gate (B) controls the tunnel coupling between the dots, allowing the precise tuning of the exchange coupling from <100 kHz to 20 MHz. To minimize the sensitivity to charge noise, we activate the

[1]QuTech, Delft University of Technology, Delft, The Netherlands. [2]Kavli Institute of Nanoscience, Delft University of Technology, Delft, The Netherlands. [3]Netherlands Organisation for Applied Scientific Research (TNO), Delft, The Netherlands. ✉e-mail: l.m.k.vandersypen@tudelft.nl

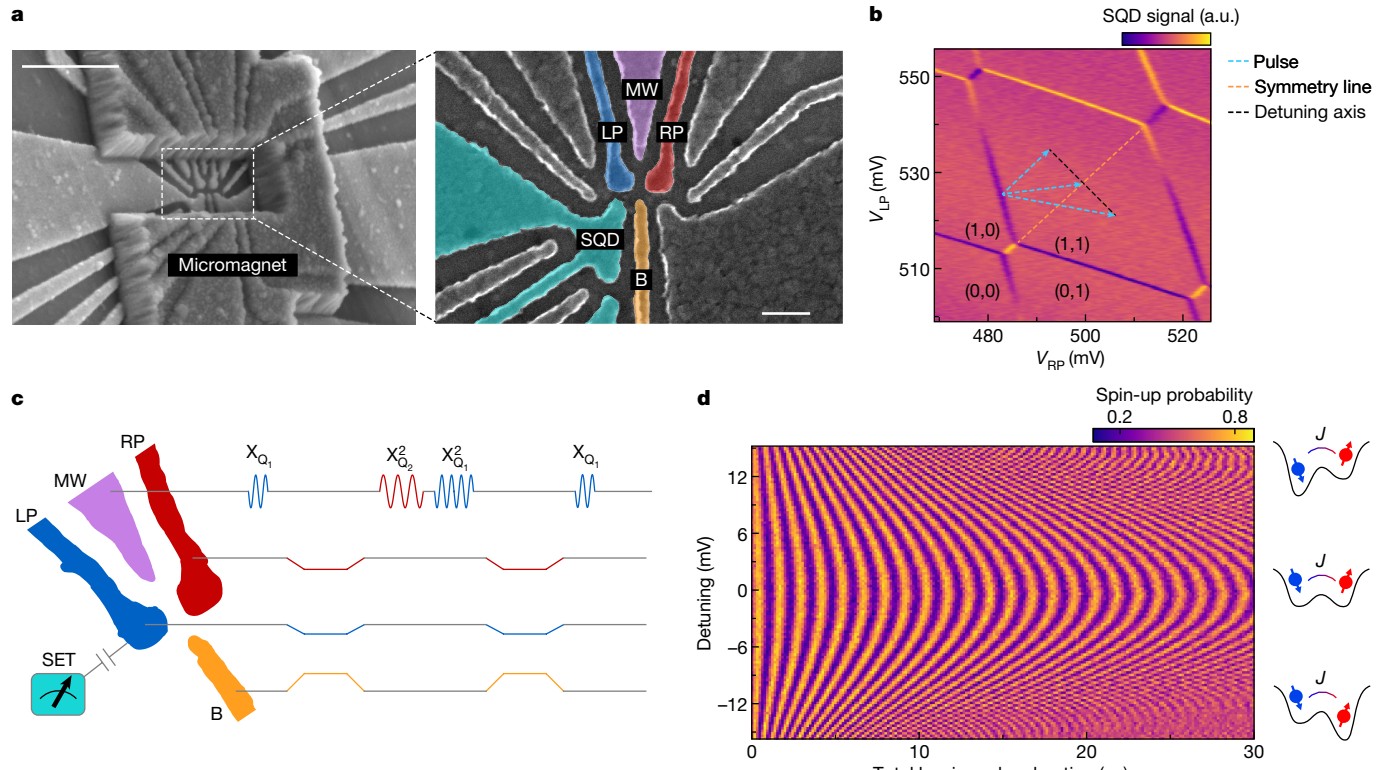

**Fig. 1 | Two-qubit device and symmetry operating point. a**, Scanning electron microscopy images of a device similar to that used here, showing the quantum dot gate pattern and the micromagnet on top (the device used in the experiment has an additional screening gate above the fine gates[17]). The scale bar in the left panel denotes 500 nm. The scale bar in the right panel denotes 100 nm. **b**, Control paths for determining the symmetry operation point in the charge-stability diagram. (*M*, *N*) represent the number of electrons in the dots underneath the tip of LP and RP, respectively. a.u., arbitrary units. **c**, Pulse sequence schematic of a decoupled controlled-phase operation interleaved in a Ramsey interference sequence on $Q_1$. **d**, Spin-up probability of $Q_1$ after the Ramsey sequence in **c**, as a function of the detuning in the double-dot potential and the total duration of the barrier voltage pulses.

exchange coupling while avoiding a tilt in the double-dot potential[24,25] (Fig. 1a). This symmetric condition can be determined accurately by decoupled adiabatic exchange pulses inside a Ramsey sequence (Fig. 1c, d). The tunnel barrier is controlled by simultaneously pulsing gate B and compensating LP and RP to avoid shifts in the electrochemical potentials[24], constituting a virtual barrier gate. The detuning between quantum dots is controlled by additional offsets to the LP and RP pulses in opposite directions. As the decoupling pulses remove additional single-qubit phase accumulation from electron movement in the magnetic field gradient, the spin-up probability of $Q_1$ results in a symmetric chevron pattern, with the symmetry point at the centre (Fig. 1d).

Among the various quantum benchmarking techniques, quantum process tomography (QPT) is designed to reconstruct all details in a target process[6]. Owing to the susceptibility of QPT to state preparation and measurement (SPAM) errors, self-consistent benchmarking techniques such as GST[26] and alternative techniques such as randomized benchmarking[27] have been developed. In contrast to randomized benchmarking, GST inherits the advantage of QPT in that it reports the detailed process, which allows us to isolate Hamiltonian errors from stochastic errors and to correct for such errors in the control signals (Extended Data Fig. 5). In addition, GST accounts for gate-dependent errors. We benchmark the fidelities of a universal gate set using GST[26,28] (Fig. 2a). The gate set we choose contains an idle gate (I), sequentially operated single-qubit π/2 rotations about the $\hat{x}$ and $\hat{y}$ axes for each qubit ($X_{Q_1}$, $Y_{Q_1}$, $X_{Q_2}$ and $Y_{Q_2}$) and a two-qubit controlled-phase (CZ) gate. A total of 36 fiducial sequences containing {null, $X_{Q_i}^{n=1,2,3}$, $Y_{Q_j}^{n=1,3}$} on each qubit, where null (unlike the idle gate) has no waiting time, are used to tomographically measure the two-qubit state. These fiducials are interleaved by germ sequences and their powers up to a sequence depth of 16. Germs are short sequences of gates taken from the universal gate set

(see Methods). They are repetitively executed to amplify different types of gate errors in the gate set, such that SPAM errors can be isolated. GST allows using a maximum-likelihood estimator to compute completely positive and trace-preserving process matrices for each element of the gate set[6]. The gate fidelity can be calculated by comparing the measured process using the Pauli transfer matrix (PTM), $\mathcal{M}_{exp}$, with the ideal PTM, $\mathcal{M}_{ideal}$, $F_{gate} = (\mathrm{Tr}(\mathcal{M}_{exp}^{-1}\mathcal{M}_{ideal}) + d)/[d(d+1)]$, where $d$ is the dimension of the Hilbert space. These process matrices provide a detailed error diagnosis of the gate set, allowing for efficient feedback calibration[29] (Fig. 2a). Analysing the error generator $\mathcal{L} = \log(\mathcal{M}_{exp}\mathcal{M}_{ideal}^{-1})$ provides easy access to information. For example, coherent Hamiltonian errors can be isolated from incoherent stochastic errors and single-qubit errors can be isolated from each other and from two-qubit errors[30].

Figure 2b, c shows the reduced PTMs of $X_{Q_1}$ and $Y_{Q_1}$ operations in the $Q_1$ subspace and Fig. 2d shows the full PTM of $Y_{Q_1}$ in two-qubit space ($Y_{Q_1} \otimes I_{Q_2}$) containing additional errors from decoherence and crosstalk on $Q_2$ while operating $Q_1$ (see Extended Data Figs. 1 and 2 for other PTMs) and from unintentional entanglement due to a residual exchange interaction. The average single-qubit gate fidelity is 99.72% in the single-qubit subspace ($X_{Q_1}$: 99.68%; $Y_{Q_1}$: 99.73%; $X_{Q_2}$: 99.61%; $Y_{Q_2}$: 99.87%; see Extended Data Fig. 2 for all error bars). A metric that is rarely reported is the single-qubit gate fidelity in the full two-qubit space, here 99.16% on average (see Methods and Extended Data Fig. 1). These results highlight that single-qubit benchmarking is not sufficient to identify all errors occurring during single-qubit operations. By analysing the error generators, we find that errors from uncorrelated dephasing of the idling qubit dominate the drop in single-qubit gate fidelity when characterized in the two-qubit space. Coherent, microwave-induced phase shifts—the main source of crosstalk errors—have been corrected by applying a compensating phase gate to the idling qubit (Extended Data Fig. 4). The elimination of idling errors and other crosstalk

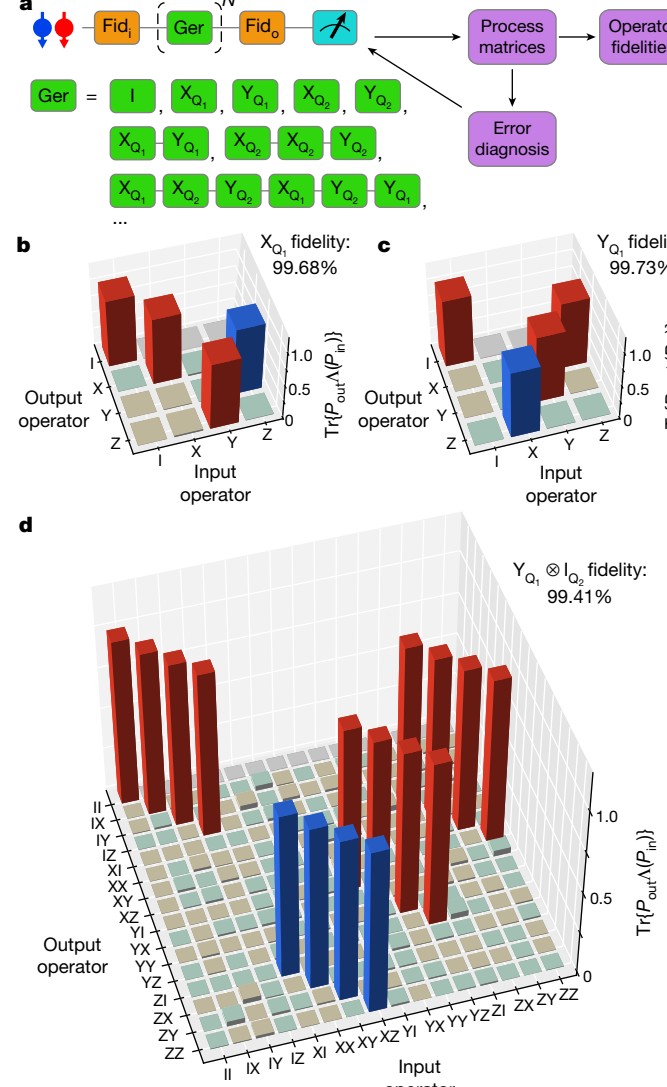

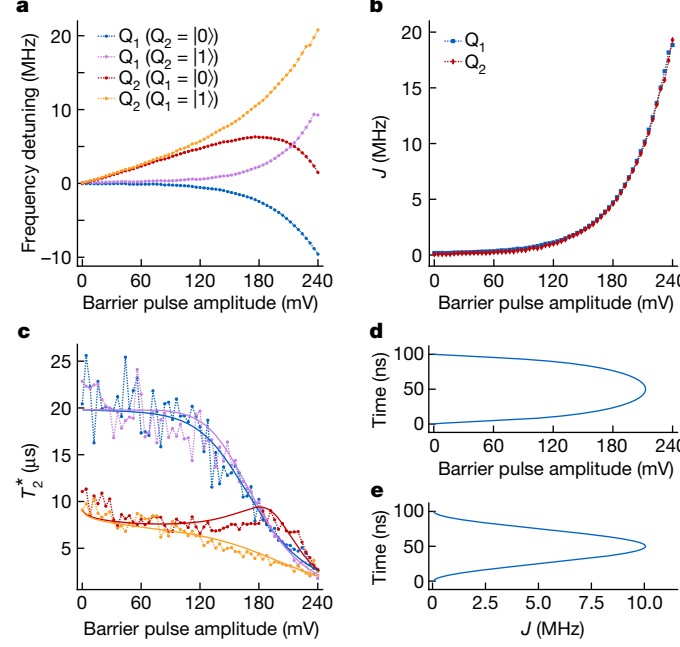

**Fig. 2 | Gate-set tomography and single-qubit gate. a**, Workflow of the GST experiment. Coloured blocks show the input and output fiducial sequences (Fid$_i$ and Fid$_o$, orange) and the germ sequences (green). A few examples of single-qubit germ sequences are listed. The outcome is used to adjust pulse parameters in the next run. **b**, **c**, PTMs of $X_{Q_1}$ and $Y_{Q_1}$ in the subspace of $Q_1$. The red (blue) bars are theoretically +1 (−1) and are measured to be positive (negative). The brown (green) bars are theoretically 0 (0) but measured to be positive (negative). $P_{in}$ and $P_{out}$ are the input and output operators, respectively. **d**, Experimentally measured PTM of $Y_{Q_1} \otimes I_{Q_2}$ in the complete two-qubit space. The colour code is the same as in **b**, **c**.

errors from the microwave drive, such as through heating effects, will be a crucial step to improve the quality of the single-qubit operations further.

For a high-fidelity adiabatic CZ gate, precise control of the exchange coupling, $J$, between the two qubits is required. Specifically, in order to avoid unintended state transitions due to non-adiabatic dynamics, we must be able to carefully shape the envelope of $J$. We characterize $J$ over a wide range using a Ramsey sequence interleaved by a virtual barrier pulse with incremental amplitude $\nu_B$. Figure 3a shows the measured frequency shift of each qubit as functions of the barrier pulse amplitude and the state of the other qubit. The exchange interaction is modelled to be exponentially dependent on the barrier pulse amplitude $J(\nu_B) \propto e^{2\alpha\nu_B}$ (refs. [31,32]). The micromagnet-induced single-qubit frequency shifts are approximated by linear functions within the voltage window of the CZ gate in the numerical simulations. By fitting the measured datasets simultaneously to theoretical models (see Methods),

**Fig. 3 | Hamiltonian engineering of exchange interaction. a**, Frequency detuning of each qubit conditional on the state of the other qubit as a function of barrier pulse amplitude. The horizontal axis shows the real voltage applied to gate B. **b**, Exchange strength as a function of barrier pulse amplitude. The data are extracted directly from **a**. **c**, $T_2^*$ of each qubit conditional on the state of the other qubit as a function of barrier pulse amplitude (same colour code as in **a**). Each data point is averaged for about 8 min. By fitting the $T_2^*$ values to a quasistatic noise model (solid lines, see Methods), the low-frequency amplitudes of the fluctuations are estimated as $\delta f_{Q_1} = 11$ kHz, $\delta f_{Q_2} = 24$ kHz and $\delta\nu_B = 0.4$ mV. **d**, Shape of the barrier pulse, designed to achieve a high-fidelity CZ gate. **e**, The cosine-shaped $J$ envelope seen by the qubits during the pulse shown in **d**.

$J$ can be extracted very precisely as the difference between the two conditional frequencies of each qubit[16,33] (Fig. 3b). The barrier pulse $\nu_B \propto \log(A_{\nu_B}(1 - \cos(2\pi t/t_{gate}))/2)$ (Fig. 3d) compensates the exponential dependence such that $J \propto (1 - \cos(2\pi t/t_{gate}))$ follows a cosine window function, which ensures good adiabaticity[34] (Fig. 3e). In addition, the virtual gates are calibrated such that the symmetric operation point is maintained for each barrier setting, minimizing the influence of charge noise via the double-dot detuning. The most relevant remaining noise sources include charge noise, affecting $J$ through fluctuations in the virtual barrier gate $\delta\nu_B$, and fluctuating qubit frequencies $\delta f_{Q_1}$ and $\delta f_{Q_2}$ from charge noise entering through artificial spin–orbit coupling from the micromagnet and residual nuclear spin noise coupling through the hyperfine interaction. By analysing the decay of the Ramsey oscillations at each transition frequency, individual dephasing times $T_2^*$ can be extracted and, from there, also $\delta\nu_B$, $\delta f_{Q_1}$ and $\delta f_{Q_2}$ (Fig. 3c).

Figure 4a shows an example GST pulse sequence that contains twice in a row the germ $[CZ, X_{Q_2}, Y_{Q_1}, CZ, Y_{Q_2}, X_{Q_1}]$. The PTM of the CZ gate obtained from GST is shown in Fig. 4b. Using the detailed information from the error generator to fine-tune the calibration parameters, we can achieve a CZ fidelity of 99.65 ± 0.15% (Extended Data Figs. 4 and 5). Error bars included here and elsewhere are the $2\sigma \approx 95\%$ confidence intervals computed using the Hessian of the loglikelihood function[35]. The CZ error generator reveals that, at this point, incoherent errors dominate. The virtual barrier gate technique used here efficiently suppresses crosstalk errors during two-qubit gates. Therefore, we expect the CZ fidelity to be mostly affected by dephasing errors of idling qubits in a larger space, which can be corrected for using decoupling pulses. From the obtained PTMs, we can numerically estimate Bell-state fidelities by multiplications of the PTMs necessary to construct the corresponding state, giving an estimate of 97.75%–98.42%, neglecting SPAM errors, for the four Bell states (Fig. 4c and Extended Data Fig. 3).

**a**

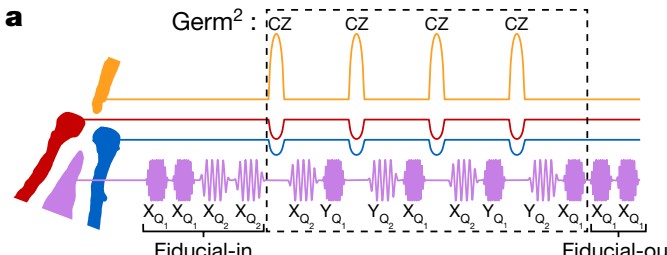

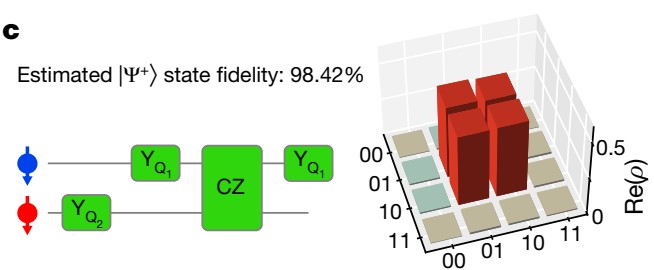

**c**

Estimated $|\Psi^+\rangle$ state fidelity: 98.42%

**b**

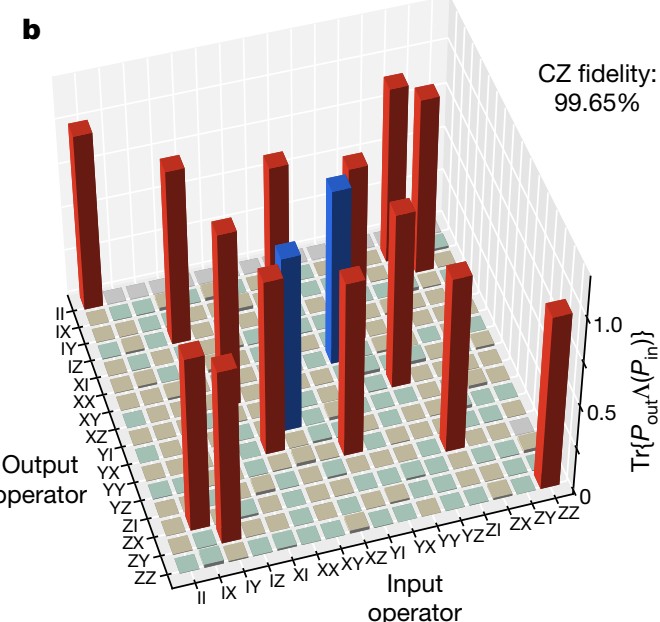

CZ fidelity: 99.65%

**Fig. 4 | High-fidelity two-qubit gate. a**, A sequence of pulses generated by the arbitrary waveform generators in an example GST sequence. The purple waveforms show the in-phase component of X/Y gates. The CZ gate is indicated by the orange pulse of gate B and the blue and red compensation pulses of gate LP and gate RP. **b**, Experimentally determined PTM of a CZ gate. The colour code is the same as in Fig. 2. **c**, Left, the quantum circuit used to reconstruct the Bell state $|\Psi^+\rangle = (|01\rangle + |10\rangle)/\sqrt{2}$ based on the corresponding PTMs. Right, the real part of the reconstructed density matrix of the $|\Psi^+\rangle$ state. The colour code is the same as in Fig. 2, except that red (blue) bars here are theoretically +0.5 (−0.5).

Next, we use the high-fidelity gate set in the context of an actual application, in order to provide a quantitative benchmark for future work under realistic conditions. Specifically, we implement a VQE algorithm to compute the ground-state energy of molecular hydrogen ($H_2$) (Fig. 5a). In a VQE algorithm, a quantum processor is used to implement a classically inefficient subroutine (see Methods and Extended Data Fig. 6). The second quantized $H_2$ Hamiltonian can be mapped onto two qubits under the Bravyi–Kitaev (BK) transformation $H = h_0 II + h_1 ZI + h_2 IZ + h_3 ZZ + h_4 XX + h_5 YY$. Here I, X, Y and Z are Pauli operators, for example, ZI is shorthand for $Z \otimes I$, and the coefficients $h_0-h_5$ are classically computable functions of the internuclear distance, $R$. Figure 5b shows the schematic of the VQE algorithm and its circuit implementation for a $H_2$ molecule. The qubit is initialized in $|01\rangle$, which represents double occupation of the lowest molecular orbital, corresponding to the Hartree–Fock (HF) ground state. A parameterized ansatz state is then prepared by considering single and double excitation, which, after the BK transformation, yields $|\psi(\theta)\rangle = e^{-i\theta XY}|01\rangle$, with $\theta$ the parameter to variationally optimize. By performing partial tomography on the ansatz state with an initial guess $\theta_0$, the expectation value of the Hamiltonian for $|\psi(\theta_0)\rangle$ can be calculated. A classical computer can efficiently compute the next guess $\theta_1$ as the new input for the quantum computer. This loop is iterated until the result converges. For a $H_2$ molecule, there is only one parameter $\theta$ to optimize, thus, a scan of the entire parameter range of $2\pi$ with finite samples is sufficient to interpolate the smoothly changing measured expectation values. This emulates a real variational algorithm, where $\theta$ can be estimated to arbitrary precision by increasing the number of repetitions to suppress statistical fluctuations[36]. Figure 5c shows the partial tomography result after normalization of the visibility window. The data demonstrate high-quality phase control in the quantum circuits. The deviations in the odd-parity expectation values indicate correlations in the readout of the two qubits[37]. Figure 5d shows the energy curves of the $H_2$ molecule from both theory[38] and the VQE experiment. We observe a minimum energy at around 0.72 Å and an error of approximately 20 mHa at the theoretical bond length 0.7414 Å, mainly attributed to slow drift in the readout parameters. This accuracy matches the results obtained using superconducting and trapped ion qubits with comparable gate fidelities[36,39].

The two-qubit gate with fidelity above 99.5% and single-qubit gate fidelities in the two-qubit gate space above 99% on average place semiconductor spin qubit logic at the error threshold of the surface code. Recently, a two-qubit operation between nuclear spin qubits in silicon, mediated by an electron spin qubit, has been demonstrated to surpass 99% fidelity as well, further highlighting that semiconductor spin qubits offer precise two-qubit logic[40]. Independent studies have shown spin qubit readout with a fidelity above 98% in only a few μs (ref. [41]), with further improvements underway[42]. Combining high-fidelity initialization, readout and control into a demonstration of fault tolerance poses several key challenges to be overcome. First, sufficiently large and reliable quantum dot arrays must be constructed, with good connectivity between the qubits. Second, the fidelities achieved in small-scale systems must be maintained across such larger systems, which will require reducing idling and crosstalk errors. The same advances will allow us to implement more sophisticated algorithms in the noisy intermediate-scale quantum era, such as solving energies involving excited states of more complex molecules.

## Online content

1. Lidar, D. A. & Brun, T. A. *Quantum Error Correction* (Cambridge Univ. Press, 2013).
2. Raussendorf, R. & Harrington, J. Fault-tolerant quantum computation with high threshold in two dimensions. *Phys. Rev. Lett.* **98**, 190504 (2007).
3. Fowler, A. G., Mariantoni, M., Martinis, J. M. & Cleland, A. N. Surface codes: towards practical large-scale quantum computation. *Phys. Rev. A* **86**, 032324 (2012).
4. Zwerver, A. M. J. et al. Qubits made by advanced semiconductor manufacturing. Preprint at https://arxiv.org/abs/2101.12650 (2021).
5. McArdle, S., Endo, S., Aspuru-Guzik, A., Benjamin, S. C. & Yuan, X. Quantum computational chemistry. *Rev. Mod. Phys.* **92**, 015003 (2020).
6. Nielsen, M. A. & Chuang, I. *Quantum Computation and Quantum Information* (Cambridge Univ. Press, 2002).
7. Preskill, J. Quantum computing in the NISQ era and beyond. *Quantum* **2**, 79 (2018).
8. Veldhorst, M. et al. A two-qubit logic gate in silicon. *Nature* **526**, 410–414 (2015).
9. Zajac, D. M., Hazard, T. M., Mi, X., Nielsen, E. & Petta, J. R. Scalable gate architecture for a one-dimensional array of semiconductor spin qubits. *Phys. Rev. Appl.* **6**, 054013 (2016).

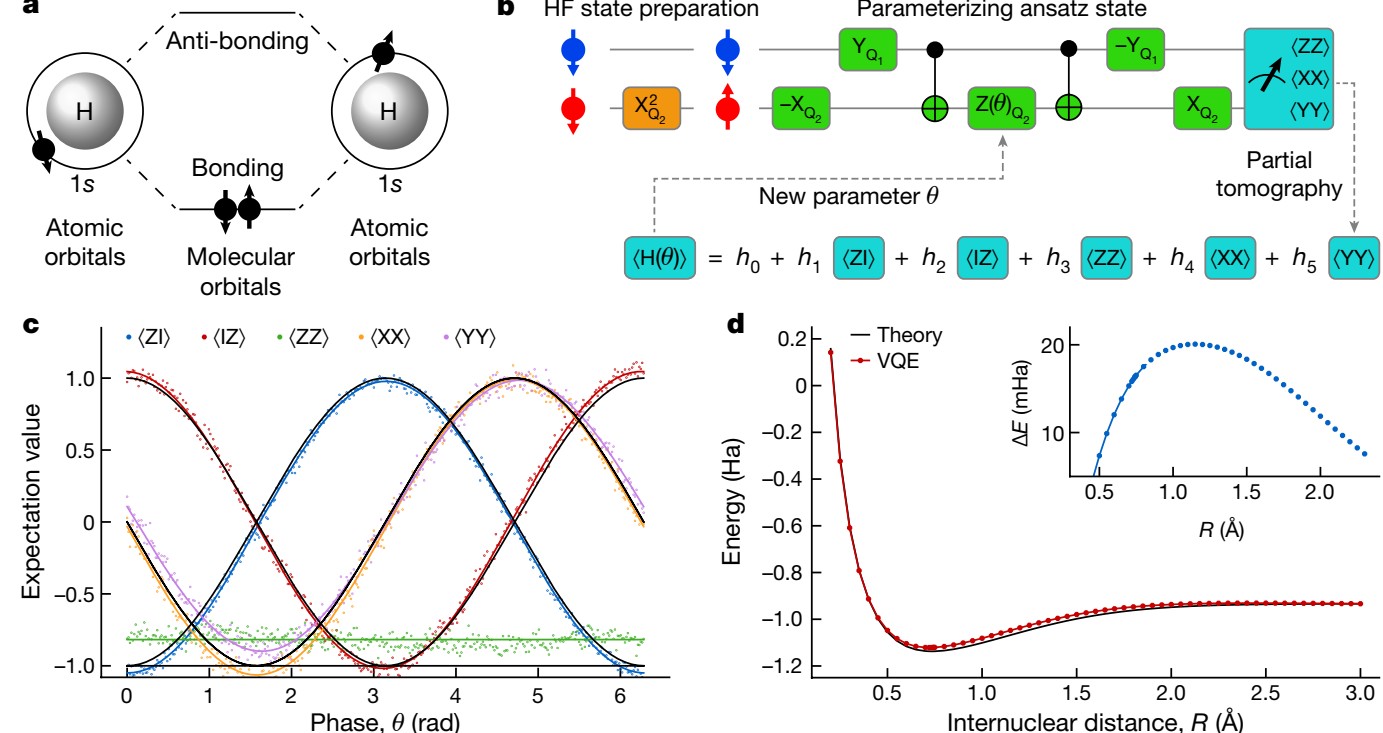

**Fig. 5 | Variational quantum eigensolver. a,** Lowest two molecular orbitals of a $H_2$ molecule, formed by the $1s$ orbitals of two hydrogen atoms. **b,** The quantum circuit to implement the VQE algorithm for a $H_2$ molecule. The orange block prepares the HF initial state by flipping $Q_2$. The circuit in green blocks creates the parameterized ansatz state. $-X_{Q_i}$ and $-Y_{Q_j}$ include virtual Z gates. CNOT gates are compiled as $[-Y_{Q_2}, CZ, Y_{Q_2}]$. To make use of the high-fidelity CZ gate, such compilation is preferred instead of using a single controlled-phase gate with

incremental length for creating the parameterized ansatz state. **c,** Expectation values of the operators in the two-qubit Hamiltonian under BK transformation as a function of $\theta$. Black solid lines show the predicted values. The coloured solid lines are sinusoidal fits to the data (and a constant fit for the case of ZZ). **d,** Potential energy of the $H_2$ molecule at varying $R$. The VQE data are normalized to the theoretical energy at large $R$ to directly compare the dissociation energy with the theoretical value. The inset shows the error in the normalized experimental data.

10.  Vandersypen, L. M. K. et al. Interfacing spin qubits in quantum dots and donors–hot, dense, and coherent. *npj Quantum Inf.* **3**, 34 (2017).
11.  Yoneda, J. et al. A quantum-dot spin qubit with coherence limited by charge noise and fidelity higher than 99.9%. *Nat. Nanotechnol.* **13**, 102–106 (2018).
12.  Yang, C. H. et al. Silicon qubit fidelities approaching incoherent noise limits via pulse engineering. *Nat. Electron.* **2**, 151–158 (2019).
13.  Xue, X. et al. Benchmarking gate fidelities in a Si/SiGe two-qubit device. *Phys. Rev. X* **9**, 021011 (2019).
14.  Huang, W. et al. Fidelity benchmarks for two-qubit gates in silicon. *Nature* **569**, 532–536 (2019).
15.  Takeda, K. et al. Quantum tomography of an entangled three-qubit state in silicon. *Nat. Nanotechnol.* **16**, 965–969 (2021).
16.  Watson, T. F. et al. A programmable two-qubit quantum processor in silicon. *Nature* **555**, 633–637 (2018).
17.  Xue, X. et al. CMOS-based cryogenic control of silicon quantum circuits. *Nature* **593**, 205–210 (2021).
18.  Elzerman, J. M. et al. Single-shot read-out of an individual electron spin in a quantum dot. *Nature* **430**, 431–435 (2004).
19.  Pioro-Ladrière, M. et al. Electrically driven single-electron spin resonance in a slanting Zeeman field. *Nat. Phys.* **4**, 776–779 (2008).
20.  Vandersypen, L. M. K. & Chuang, I. L. NMR techniques for quantum control and computation. *Rev. Mod. Phys.* **76**, 1037 (2005).
21.  Loss, D. & DiVincenzo, D. P. Quantum computation with quantum dots. *Phys. Rev. A* **57**, 120 (1998).
22.  Petta, J. R. et al. Coherent manipulation of coupled electron spins in semiconductor quantum dots. *Science* **309**, 2180–2184 (2005).
23.  Meunier, T., Calado, V. E. & Vandersypen, L. M. K. Efficient controlled-phase gate for single-spin qubits in quantum dots. *Phys. Rev. B* **83**, 121403 (2011).
24.  Martins, F. et al. Noise suppression using symmetric exchange gates in spin qubits. *Phys. Rev. Lett.* **116**, 116801 (2016).
25.  Reed, M. D. et al. Reduced sensitivity to charge noise in semiconductor spin qubits via symmetric operation. *Phys. Rev. Lett.* **116**, 110402 (2016).
26.  Blume-Kohout, R. et al. Demonstration of qubit operations below a rigorous fault tolerance threshold with gate set tomography. *Nat. Commun.* **8**, 14485 (2017).
27.  Magesan, E., Gambetta, J. M. & Emerson, J. Characterizing quantum gates via randomized benchmarking. *Phys. Rev. A* **85**, 042311 (2012).
28.  Dehollain, J. P. et al. Optimization of a solid-state electron spin qubit using gate set tomography. *New J. Phys.* **18**, 103018 (2016).
29.  White, G. A., Hill, C. D. & Hollenberg, L. C. Performance optimization for drift-robust fidelity improvement of two-qubit gates. *Phys. Rev. Appl.* **15**, 014023 (2021).
30.  Blume-Kohout, R. et al. A taxonomy of small Markovian errors. Preprint at https://arxiv.org/abs/2103.01928 (2021).
31.  Cerfontaine, P., Otten, R., Wolfe, M. A., Bethke, P. & Bluhm, H. High-fidelity gate set for exchange-coupled singlet-triplet qubits. *Phys. Rev. B* **101**, 155311 (2020).
32.  Pan, A. et al. Resonant exchange operation in triple-quantum-dot qubits for spin–photon transduction. *Quantum Sci. Technol.* **5**, 034005 (2020).
33.  Zajac, D. M. et al. Resonantly driven CNOT gate for electron spins. *Science* **359**, 439–442 (2017).
34.  Martinis, J. M. & Geller, M. R. Fast adiabatic qubit gates using only $\sigma_z$ control. *Phys. Rev. A* **90**, 022307 (2014).
35.  Nielsen, E. et al. Gate set tomography. *Quantum* **5**, 557 (2021).
36.  Hempel, C. et al. Quantum chemistry calculations on a trapped-ion quantum simulator. *Phys. Rev. X* **8**, 031022 (2018).
37.  Chow, J. M. et al. Detecting highly entangled states with a joint qubit readout. *Phys. Rev. A* **81**, 062325 (2010).
38.  McClean, J. R. et al. OpenFermion: the electronic structure package for quantum computers. *Quantum Sci. Technol.* **5**, 034014 (2020).
39.  Ganzhorn, M. et al. Gate-efficient simulation of molecular eigenstates on a quantum computer. *Phys. Rev. Appl.* **11**, 044092 (2019).
40.  Mądzik, M. T. et al. Precision tomography of a three-qubit donor quantum processor in silicon. *Nature* **601**, 348–353 (2022).
41.  Zheng, G. et al. Rapid gate-based spin read-out in silicon using an on-chip resonator. *Nat. Nanotechnol.* **14**, 742–746 (2019).
42.  Schaal, S. et al. Fast gate-based readout of silicon quantum dots using Josephson parametric amplification. *Phys. Rev. Lett.* **124**, 067701 (2020).

## Methods

### Measurement setup

The measurement setup and device are similar to those used in ref. [17]. We summarize a few key points and all the differences here. The gates LP, RP and B are connected to arbitrary waveform generators (AWGs, Tektronix 5014C) via coaxial cables. The position in the charge-stability diagram of the quantum dots is controlled by voltage pulses applied to LP and RP. Linear combinations of the voltage pulses applied to B, LP and RP are used to control the exchange coupling between the two qubits at the symmetry point. The compensation coefficients are $v_{LP}/v_B = -0.081$ and $v_{RP}/v_B = 0.104$. A vector signal generator (VSG, Keysight E8267D) is connected to gate MW and sends frequency-multiplexed microwave bursts (not necessarily time-multiplexed) to implement electric-dipole spin resonance (EDSR). The VSG has two I/Q input channels, receiving I/Q modulation pulses from two channels of an AWG. I/Q modulation is used to control the frequency, phase and length of the microwave bursts. The current signal of the sensing quantum dot is converted to a voltage signal and recorded by a digitizer card (Spectrum M4i.44), and then converted into 0 or 1 by comparing it to a threshold value.

Two differences between the present setup and that in ref. [17] are that (1) the programmable mechanical switch is configured such that gate MW is always connected to the VSG and not to the cryo-CMOS control chip and (2) a second AWG of the same model is connected to gate B, with its clock synchronized to the first AWG.

### Gate calibration

In the gate set used in this work, {I, $X_{Q_1}$, $Y_{Q_1}$, $X_{Q_2}$, $Y_{Q_2}$, CZ}, the duration of the I gate and the CZ gate are set to 100 ns, and we calibrate and keep the amplitudes of the single-qubit drives fixed and in the linear-response regime, where the Rabi frequency is linearly dependent on the driving amplitude. The envelopes of the single-qubit gates are shaped following a 'Tukey' window, as it allows adiabatic single-qubit gates with relatively small amplitudes, thus, avoiding the distortion caused by a nonlinear response. The general Tukey window of length $t_p$ is given by

$$W(t,r) = \begin{cases} \frac{1}{2}\left[1 - \cos\left(\frac{2\pi t}{rt_p}\right)\right] & 0 \le t \le \frac{rt_p}{2} \\ 1 & \frac{rt_p}{2} < t < t_p - \frac{rt_p}{2}, \\ \frac{1}{2}\left[1 - \cos\left(\frac{2\pi(t_p - t)}{rt_p}\right)\right] & t_p - \frac{rt_p}{2} \le t \le t_p \end{cases} \quad (1)$$

where $r = 0.5$ for our pulses. Apart from these fixed parameters, there are 11 free parameters that must be calibrated: single-qubit frequencies $f_{Q_1}$ and $f_{Q_2}$, burst lengths for single-qubit gates $t_{XY1}$ and $t_{XY2}$, phase shifts caused by single-qubit gates on the addressed qubit itself $\phi_{11}$ and $\phi_{22}$, phase shifts caused by single-qubit gates on the unaddressed 'victim qubit' $\phi_{12}$ and $\phi_{21}$ ($\phi_{12}$ is the phase shift on $Q_1$ induced by a gate on $Q_2$ and similar for $\phi_{21}$), the peak amplitude of the CZ gate $A_{v_B}$ and phase shifts caused by the gate voltage pulses used for the CZ gate on the qubits $\theta_1$ and $\theta_2$ (in addition, we absorb into $\theta_1$ and $\theta_2$ the 90° phase shifts needed to transform diag(1, i, i, 1) into diag(1, 1, 1, −1)).

For single-qubit gates, $f_{Q_1}$ and $f_{Q_2}$ are calibrated by standard Ramsey sequences, which are automatically executed every 2 h, at the beginning and in the middle (after 100 times the average of each sequence) of the GST experiment. The EDSR burst times $t_{XY1}$ and $t_{XY2}$ are initially calibrated by an AllXY calibration protocol [43]. The phases $\phi_{11}$, $\phi_{12}$, $\phi_{21}$ and $\phi_{22}$ are initially calibrated by measuring the phase shift of the victim qubit ($Q_1$ for $\phi_{11}$ and $\phi_{21}$; $Q_2$ for $\phi_{22}$ and $\phi_{12}$) in a Ramsey sequence interleaved by a pair of $[X_{Q_j} - X_{Q_j}]$ gates on the addressed qubit ($Q_1$ for $\phi_{11}$ and $\phi_{12}$; $Q_2$ for $\phi_{22}$ and $\phi_{21}$) (Extended Data Fig. 4).

The optimal pulse design presented in Fig. 3 gives a rough guidance of the pulse amplitude $A_{v_B}$. In a more precise calibration of the CZ gate, an optional π-rotation is applied to the control qubit (for example, $Q_1$)

to prepare it into the $|0\rangle$ or $|1\rangle$ state, followed by a Ramsey sequence on the target qubit ($Q_2$) interleaved by an exchange pulse. The amplitude is precisely tuned to bring $Q_2$ completely out of phase (by 180°) between the two measurements (Extended Data Fig. 4d, e). The phase $\theta_2$ is determined such that the phase of $Q_2$ changes by zero (π) when $Q_1$ is in the state $|0\rangle$ ($|1\rangle$), corresponding to CZ = diag(1, 1, 1, −1) in the standard basis. The same measurement is then performed again with $Q_2$ as the control qubit and $Q_1$ as the target qubit to determine $\theta_1$ (ref. [16]).

In such a 'conventional' calibration procedure of the CZ gate, we notice that the two qubits experience different conditional phases (Extended Data Fig. 4). We believe that this effect is caused by off-resonant driving from the optional π-rotation on the control qubit. Similar effects can also affect the calibration of the phase crosstalk from single-qubit gates.

This motivates us to use the results from GST as feedback to adjust the gate parameters. The error generators not only describe the total errors of the gates but also distinguish Hamiltonian errors (coherent errors) from stochastic errors (incoherent errors). We use the information on seven different Hamiltonian errors (IX, IY, XI, YI, ZI, IZ and ZZ) of each gate to correct all 11 gate parameters (Extended Data Fig. 5), except $f_{Q_1}$ and $f_{Q_2}$, for which calibrations using standard Ramsey sequences are sufficient. For single-qubit gates, $t_{XY1}$ and $t_{XY2}$ are adjusted according to the IX, IY, XI and YI errors. The phases $\phi_{11}$, $\phi_{12}$, $\phi_{21}$ and $\phi_{22}$ are adjusted according to the ZI and IZ errors. For the CZ gate, $\theta_1$ and $\theta_2$ are adjusted according to the ZI and IZ errors, and $A_{v_B}$ is adjusted according to the ZZ error. The adjusted gates are then used in a new GST experiment.

### Theoretical model

In this section, we describe the theoretical model used for the fitting, the pulse optimization and the numerical simulations. The dynamics of two electron spins in the (1,1) charge configuration can be accurately described by an extended Heisenberg model [21]

$$H = g\mu_B \mathbf{B}_1 \cdot \mathbf{S}_1 + g\mu_B \mathbf{B}_2 \cdot \mathbf{S}_2 + hJ\left(\mathbf{S}_1 \cdot \mathbf{S}_2 - \frac{1}{4}\right), \quad (2)$$

with $\mathbf{S}_j = (X_j, Y_j, Z_j)^T/2$, where $X_j$, $Y_j$ and $Z_j$ are the single-qubit Pauli matrices acting on spin $j = 1, 2$, $\mu_B$ the Bohr's magneton, $g \approx 2$ the $g$-factor in silicon and $h$ is Planck's constant. The first and second terms describe the interaction of the electron spin in dot 1 and dot 2 with the magnetic fields $\mathbf{B}_j = (B_{x,j}, 0, B_{z,j})^T$ originating from the externally applied field and the micromagnet. The transverse components $B_{x,j}$ induce spin-flips, thus, single-qubit gates if modulated resonantly via EDSR. For later convenience, we define the resonance frequencies by $hf_{Q_1} = g\mu_B B_{z,1}$ and $hf_{Q_2} = g\mu_B B_{z,2}$, and the energy difference between the qubits $\Delta E_z = g\mu_B(B_{z,2} - B_{z,1})$. The last term in the Hamiltonian of equation (2) describes the exchange interaction $J$ between the spins in neighbouring dots. The exchange interaction originates from the overlap of the wave functions through virtual tunnelling events and is, in general, a nonlinear function of the applied barrier voltage $v_B$. We note that $v_B$ determines the compensation pulses applied to LP and RP for virtual barrier control. We model $J$ as an exponential function [31,32]

$$J(v_B) = J_{res} e^{2\alpha v_B}, \quad (3)$$

where $J_{res} \approx 20$–100 kHz is the residual exchange interaction during idle and single-qubit operations and $\alpha$ is the lever arm. In general, the magnetic fields $\mathbf{B}_j$ depend on the exact position of the electron. We include this in our model $B_{z,j} \to B_{z,j}(v_B) = B_{z,j}(0) + \beta_j v_B^{\gamma}$, where $\beta_j$ accounts for the impact of the barrier voltage on the resonance frequency of qubit $j$. The transition energies described in the main text are now given by diagonalizing the Hamiltonian from equation (2) and computing the energy difference between the eigenstates corresponding to the computational basis states {$|00\rangle$, $|01\rangle$, $|10\rangle$, $|11\rangle$} (ref. [44]). We have

$$hf_{Q_1}(Q_2 = |0\rangle) = \mathcal{E}(|10\rangle) - \mathcal{E}(|00\rangle), \tag{4}$$

$$hf_{Q_1}(Q_2 = |1\rangle) = \mathcal{E}(|11\rangle) - \mathcal{E}(|01\rangle), \tag{5}$$

$$hf_{Q_2}(Q_1 = |0\rangle) = \mathcal{E}(|01\rangle) - \mathcal{E}(|00\rangle), \tag{6}$$

$$hf_{Q_2}(Q_1 = |1\rangle) = \mathcal{E}(|11\rangle) - \mathcal{E}(|10\rangle), \tag{7}$$

where $\mathcal{E}(|\xi\rangle)$ denotes the eigenenergy of eigenstate $|\xi\rangle$ and $|0\rangle = |\downarrow\rangle$ is defined by the magnetic field direction.

In the presence of noise, qubits start to lose information. In silicon, charge noise and nuclear noise are the dominating sources of noise. In the absence of two-qubit coupling and correlated charge noise, both qubits decohere largely independently of each other, giving rise to a decoherence time set by the interaction with the nuclear spins and charge noise coupling to the qubit via intrinsic and artificial (via the inhomogeneous magnetic field) spin–orbit interaction. We describe this effect by $f_{Q_1} \rightarrow f_{Q_1} + \delta f_{Q_1}$ and $f_{Q_2} \rightarrow f_{Q_2} + \delta f_{Q_2}$, where $\delta f_{Q_1}$ and $\delta f_{Q_2}$ are the single-qubit frequency fluctuations. Charge noise can additionally affect both qubits via correlated frequency shifts and the exchange interaction through the barrier voltage, which we model as $v_B \rightarrow v_B + \delta v_B$. In the presence of finite exchange coupling, one can define four distinct, pure dephasing times, each corresponding to the dephasing of a single qubit with the other qubit in a specific basis state. In a quasistatic approximation, the four dephasing times are then given by

$$T_2^*(Q_1(Q_2 = |0\rangle))$$
$$= \frac{1}{\sqrt{2}\,\pi \sqrt{\left[\frac{d\left(hf_{Q_1}(Q_2 = |0\rangle)\right)}{dv_B}\right]^2 \delta v_B^2 + \left[\frac{d\left(hf_{Q_1}(Q_2 = |0\rangle)\right)}{dhf_{Q_1}}\right]^2 \delta f_{Q_1}^2}}, \tag{8}$$
$$\overline{+\left[\frac{d\left(hf_{Q_1}(Q_2 = |0\rangle)\right)}{dhf_{Q_2}}\right]^2 \delta f_{Q_2}^2}$$

$$T_2^*(Q_1(Q_2 = |1\rangle))$$
$$= \frac{1}{\sqrt{2}\,\pi \sqrt{\left[\frac{d\left(hf_{Q_1}(Q_2 = |1\rangle)\right)}{dv_B}\right]^2 \delta v_B^2 + \left[\frac{d\left(hf_{Q_1}(Q_2 = |1\rangle)\right)}{dhf_{Q_1}}\right]^2 \delta f_{Q_1}^2}}, \tag{9}$$
$$\overline{+\left[\frac{d\left(hf_{Q_1}(Q_2 = |1\rangle)\right)}{dhf_{Q_2}}\right]^2 \delta f_{Q_2}^2}$$

$$T_2^*(Q_2(Q_1 = |0\rangle))$$
$$= \frac{1}{\sqrt{2}\,\pi \sqrt{\left[\frac{d\left(hf_{Q_2}(Q_1 = |0\rangle)\right)}{dv_B}\right]^2 \delta v_B^2 + \left[\frac{d\left(hf_{Q_2}(Q_1 = |0\rangle)\right)}{dhf_{Q_1}}\right]^2 \delta f_{Q_1}^2}}, \tag{10}$$
$$\overline{+\left[\frac{d\left(hf_{Q_2}(Q_1 = |0\rangle)\right)}{dhf_{Q_2}}\right]^2 \delta f_{Q_2}^2}$$

$$T_2^*(Q_2(Q_1 = |1\rangle))$$
$$= \frac{1}{\sqrt{2}\,\pi \sqrt{\left[\frac{d\left(hf_{Q_2}(Q_1 = |1\rangle)\right)}{dv_B}\right]^2 \delta v_B^2 + \left[\frac{d\left(hf_{Q_2}(Q_1 = |1\rangle)\right)}{dhf_{Q_1}}\right]^2 \delta f_{Q_1}^2}}. \tag{11}$$
$$\overline{+\left[\frac{d\left(hf_{Q_2}(Q_1 = |1\rangle)\right)}{dhf_{Q_2}}\right]^2 \delta f_{Q_2}^2}$$

## Fitting qubit frequencies and dephasing times

The transition energies in equations (4)–(7) are fitted simultaneously to the measured results from the Ramsey experiment (see Fig. 3a). For the fitting, we use the NonLinearModelFit function from the software Mathematica with the least squares method. The best fits yield the following parameters: $\alpha = 12.1 \pm 0.05\ \mathrm{V^{-1}}$, $\beta_1 = -2.91 \pm 0.11\ \mathrm{MHz\,V^{-\gamma}}$, $\beta_2 = 67.2 \pm 0.63\ \mathrm{MHz\,V^{-\gamma}}$, $\gamma = 1.20 \pm 0.01$ and $J_{res} = 58.8 \pm 1.8\ \mathrm{kHz}$.

The dephasing times in equations (8)–(11) are fitted simultaneously to the measured results from the Ramsey experiment (see Fig. 3c) using the same method. The best fits yield the following parameters: $\delta v_B = 0.40 \pm 0.01\ \mathrm{mV}$, $\delta f_{Q_1} = 11 \pm 0.1\,\mathrm{kHz}$ and $\delta f_{Q_2} = 24 \pm 0.7\,\mathrm{kHz}$.

## Numerical simulations

For all numerical simulations, we solve the time-dependent Schrödinger equation

$$i\hbar \frac{d}{dt}|\psi(t)\rangle = H|\psi(t)\rangle \tag{12}$$

and iteratively compute the unitary propagator according to

$$U(t + \Delta t) = e^{-\frac{i}{\hbar}H(t + \Delta t)} U(t),$$

where $\hbar = h/(2\pi)$ is the reduced Planck's constant. Here $H(t + \Delta t)$ is discretized into $N$ segments of length $\Delta t$ such that $H(t)$ is constant in the time interval $[t, t + \Delta t]$. All simulations are performed in the rotating frame of the external magnetic field $(B_{z,1} + B_{z,2})/2$ and neglecting the counter-rotating terms, making the so-called rotating-wave approximation. This allows us to choose $\Delta t = 10$ ps as a sufficiently small time step.

For the noise simulations, we included classical fluctuations of $f_{Q_1} \rightarrow f_{Q_1} + \delta f_{Q_1}$, $f_{Q_2} \rightarrow f_{Q_2} + \delta f_{Q_2}$ and $v_B \rightarrow v_B + \delta v_B$. We assume the noise coupling to the resonance frequencies $\delta f_{Q_1}$ and $\delta f_{Q_2}$ to be quasistatic and assume $1/f$ noise for $v_B$, which we describe by its spectral density $S_{v_B}(\omega) = \delta v_B/\omega$, where $\omega$ is the angular frequency. To compute time traces of the fluctuation, we use the approach introduced in refs. [45,46] to generate time-correlated time traces. The fluctuations are discretized into $N$ segments with time $\Delta t$ such that $\delta v_B(t)$ is constant in the time interval $[t, t + \Delta t)$, with the same $\Delta t$ as above. Consequently, fluctuations that are faster than $f_{max} = \frac{1}{\Delta t}$ are truncated.

## CZ gate

We realize a universal CZ = diag(1, 1, 1, −1) gate by adiabatically pulsing the exchange interaction using a carefully designed pulse shape. Starting from equation (2), the full dynamics can be projected on the odd-parity space spanned by $|01\rangle$ and $|10\rangle$. The entangling exchange gate is reduced in this subspace to a global phase shift, thus, the goal is to minimize any dynamics inside the subspace. Introducing a new set of Pauli operators in this subspace $\sigma_x = |01\rangle\langle10| + |10\rangle\langle01|$, $\sigma_y = -i|01\rangle\langle10| + i|10\rangle\langle01|$ and $\sigma_z = |01\rangle\langle01| - |10\rangle\langle10|$, we find

$$H_{sub}(t) = \frac{1}{2}\left(-hJ(v_B(t)) + \Delta E_z\,\sigma_z + hJ(v_B(t))\,\sigma_x\right). \tag{14}$$

In order to investigate the adiabatic behaviour, it is convenient to switch into the adiabatic frame defined by $U_{ad} = e^{-\frac{i}{2}\tan^{-1}\left(\frac{hJ(v_B(t))}{\Delta E_z}\right)\sigma_y}$. The Hamiltonian accordingly transforms as

$$H_{ad} = U_{ad}^\dagger(t) H_{sub}(t) U_{ad}(t) - i\hbar U_{ad}^\dagger(t)\dot{U}_{ad}(t) \tag{15}$$

$$\approx \frac{1}{2}\left(-hJ(v_B(t)) + \Delta E_z\sigma_z - \frac{h^2 \dot{J}}{2\pi\Delta E_z}\sigma_y\right), \tag{16}$$

where the first term is unaffected and describes the global phase accumulation due to the exchange interaction, the second term describes

the single-qubit phase accumulations and the last term, $f(t) = h^2 J/(4\pi\Delta E_z)$, describes the diabatic deviation proportional to the derivative of the exchange pulse. From equation (15) and equation (16), we assumed a constant $\Delta E_z(t) \approx \Delta E_z$ and $hJ(t) \ll \Delta E_z$. The transition probability from state $|\uparrow\downarrow\rangle$ to $|\downarrow\uparrow\rangle$ using a pulse of length $t_p$ is then given by[34]

$$P_{|\uparrow\downarrow\rangle\to|\downarrow\uparrow\rangle} \approx \left| \int_0^{t_p} f(t) e^{-\frac{i}{\hbar}\Delta E_z t} \mathrm{d}t \right|^2 \tag{17}$$

$$\propto S_s(f(t)). \tag{18}$$

From the first to the second line, we identify the integral by the (short-timescale) Fourier transform, allowing us to describe the spin-flip error probability by the energy spectral density $S_s$ of the input signal $f(t)$. Minimizing such errors is, therefore, identical to minimizing the energy spectral density of a pulse, a well-known and solved problem from classical signal processing and statistics. Optimal shapes are commonly referred to as window functions $W(t)$ due to their property of restricting the spectral resolution of signals. A high-fidelity exchange pulse is consequently given by $J(0) = J(t_p)$ and

$$\int_0^{t_p} \mathrm{d}t J(v_B(t)) = 1/4, \tag{19}$$

while setting $J(t) = A_{v_B} W(t) J_{\text{res}}$ (ref. [34]), with a scaling factor $A_{v_B}$ that is to be determined. In this work, we have chosen the cosine window

$$W(t) = \frac{1}{2}\left[1 - \cos\left(\frac{2\pi t}{t_p}\right)\right] \tag{20}$$

from signal processing, which has a high spectral resolution. The amplitude $A_{v_B}$ follows from condition equation (19). For a pulse length of $t_p = 100$ ns and a cosine pulse shape, we find $A_{v_B} J_{\text{res}} = 10.06$ MHz. As explained in the main text, owing to the exponential voltage-exchange relation, the target pulse shape for $J(t)$ must be converted to a barrier gate pulse, following[47]

$$v_B(t) = \frac{1}{2\alpha} \log\left(A_{v_B} W(t)\right). \tag{21}$$

Our numerical simulations predict an average gate infidelity $1 - F_{\text{gate}} < 10^{-6}$ without noise and $1 - F = 0.22 \times 10^{-3}$ with the inclusion of noise through the fluctuations $\delta f_{Q_1}$, $\delta f_{Q_2}$ and $\delta v_B$, discussed in the previous section. The measured PTMs reveal much higher rates of incoherent errors, which we attribute to drifts in the barrier voltage on a timescale much longer than the timescale on which $\delta f_{Q_1}$, $\delta f_{Q_2}$ and $\delta v_B$ were determined.

## Gate-set tomography analysis

We designed a GST experiment using the gate set $\{I, X_{Q_1}, Y_{Q_1}, X_{Q_2}, Y_{Q_2}, CZ\}$, where I is a 100-ns idle gate, $X_{Q_1}(Y_{Q_1})$ and $X_{Q_2}(Y_{Q_2})$ are single-qubit $\pi/2$ gates with rotation axis $\hat{x}(\hat{y})$ on $Q_1$ and $Q_2$, with durations of 150 ns and 200 ns, respectively, and $CZ = \text{diag}(1, 1, 1, -1)$. A classic two-qubit GST experiment consists of a set of germs designed to amplify all types of error in the gate set when repeated and a set of 36 fiducials composed by the 11 elementary operations {null, $X_{Q_1}$, $X_{Q_1}X_{Q_1}$, $X_{Q_1}X_{Q_1}X_{Q_1}$, $Y_{Q_1}$, $Y_{Q_1}Y_{Q_1}Y_{Q_1}$, $X_{Q_2}$, $X_{Q_2}X_{Q_2}$, $X_{Q_2}X_{Q_2}X_{Q_2}$, $Y_{Q_2}$, $Y_{Q_2}Y_{Q_2}Y_{Q_2}$} required to carry out quantum process tomography of the germs[48]. We use a set of 16 germs {I, $X_{Q_1}$, $Y_{Q_1}$, $X_{Q_2}$, $Y_{Q_2}$, CZ, $X_{Q_1}Y_{Q_1}$, $X_{Q_2}Y_{Q_2}$, $X_{Q_1}X_{Q_1}Y_{Q_1}$, $X_{Q_2}X_{Q_2}Y_{Q_2}$, $X_{Q_2}Y_{Q_2}CZ$, $CZX_{Q_2}X_{Q_1}X_{Q_1}$, $X_{Q_1}X_{Q_2}Y_{Q_2}X_{Q_1}Y_{Q_2}Y_{Q_1}$, $X_{Q_1}Y_{Q_2}X_{Q_2}Y_{Q_1}X_{Q_2}X_{Q_2}$, $CZX_{Q_2}Y_{Q_1}CZY_{Q_2}X_{Q_1}$, $Y_{Q_1}X_{Q_1}Y_{Q_2}X_{Q_1}X_{Q_2}X_{Q_1}Y_{Q_1}Y_{Q_2}$} (ref. [35]). Note that the null gate is the instruction for doing nothing in zero time, different from the idle gate. Simple errors such as errors in the rotation angle

of a particular gate can be amplified by simply repeating the same gate. More complicated errors such as tilts in rotation axes can only be amplified by a combination of different gates. The germs and fiducials are then compiled into GST sequences, such that each sequence consists of two fiducials interleaved by a single germ or power of germs[35] (as illustrated in Fig. 2a). The GST sequences are classified by their germ powers into lengths $L = 1, 2, 4, 8, 16...$, where a sequence of length $n$ consists of $n$ gates plus the fiducial gates. We note that the sequences used in GST are shorter than the sequences involved in other methods to self-consistently estimate the gate performance, such as randomized benchmarking. As a result, GST suffers less from drift in qubit frequencies and readout windows induced by long sequences of microwave bursts.

After the execution of all sequences, a maximum-likelihood estimation is performed to estimate the process matrices of each gate in the gate set and the SPAM probabilities. We use the open source pyGSTi Python package[49,50] to perform the maximum-likelihood estimation, as well as to design an optimized GST experiment by eliminating redundant circuits and to provide statistical error bars by computing all involved Hessians. The circuit optimization allows us to perform GST with a maximum sequence length $L_{\max} = 16$ using 1,685 different sequences in total. The pyGSTi package quantifies the Markovian-model violation of the experimental data, counting the number of standard deviations exceeding their expectation values under the $\chi^2$ hypothesis[50]. This model violation is internally translated into a more accessible goodness ratio from 0 to 5, with 5 being the best[49], where we obtain a 4 out of 5 rating, indicating remarkably small deviations from expected results. The total number of standard deviations exceeding the expected results for each $L$, as well as the contribution of each sequence to this number, can be found in the pyGSTi report, along with the supporting data.

From the GST experiment, we have extracted the PTM $\mathcal{M}_{\text{exp}}$ describing each gate in our gate set $\{I, X_{Q_1}, Y_{Q_1}, X_{Q_2}, Y_{Q_2}, CZ\}$. The PTM is isomorphically related to the conventionally used $\chi$ matrix describing a quantum process. A completely positive, trace-preserving, two-qubit PTM has 240 parameters describing the process. To obtain insight into the errors of the gates in the experiment, we first compute the error in the PTM given by $E = \mathcal{M}_{\text{exp}}\mathcal{M}_{\text{ideal}}^{-1}$, where we have adapted the convention to add the error after the ideal gate. The average gate fidelity is then conveniently given by

$$F_{\text{gate}} = \frac{\text{Tr}(\mathcal{M}_{\text{exp}}^{-1}\mathcal{M}_{\text{ideal}}) + d}{d(d+1)}. \tag{22}$$

It is related to the entanglement fidelity via $1 - F_{\text{ent}} = \frac{d+1}{d}\left(1 - F_{\text{gate}}\right)$ (ref. [51]), where $d$ is the dimension of the two-qubit Hilbert space. Although the PTM $\mathcal{M}$ perfectly describes the errors, it is more intuitive to analyse the corresponding error generator $\mathcal{L} = \log(E)$ of the process[30]. The error generator $\mathcal{L}$ relates to the error PTM $E$ in a similar way as a Hamiltonian $H$ relates to a unitary operation $U = e^{-iH}$. The error generator can be separated into several blocks. A full discussion about the error generator can be found in ref. [30]. In this work, we have used the error generator to distinguish the dynamics originating from coherent Hamiltonian errors, which can be corrected by adjusting gate parameters (see Extended Data Fig. 5), and from noisy/stochastic dynamics, which cannot be corrected easily. The coherent errors can be extracted by projecting $\mathcal{L}$ onto the $4 \times 4$-dimensional Hamiltonian space $H$. In the Hilbert–Schmidt space, the Hamiltonian projection is given by[30]

$$H_{mn} = -\frac{i}{d^2}\text{Tr}\left[(P_m^T \otimes P_n^T \otimes \mathbf{1}_d - \mathbf{1}_d \otimes P_m \otimes P_n)\mathcal{L}_{\text{sup}}\right], \tag{23}$$

where $\mathcal{L}_{\text{sup}}$ is the error generator in Liouville superoperator form, $P_m \in \{I, X, Y, Z\}$ are the extended Pauli matrices with $m, n = 0, 1, 2, 3$, $\mathbf{1}_d$ is

the $d$-dimensional identity matrix and $d = 4$ is the dimension of the two-qubit Hilbert space. To improve the calibration of our gate set, we use the information of seven different Hamiltonian errors (IX, IY, XI, YI, ZI, IZ and ZZ). To estimate coherent Hamiltonian errors and incoherent stochastic errors, two new metrics are considered[30]: the Jamiołkowski probability

$$\epsilon_J(\mathcal{L}) = -\operatorname{Tr}(\rho_J(\mathcal{L})|\Psi\rangle\langle\Psi|)), \tag{24}$$

which describes the amount of incoherent error in the process, and the Jamiołkowski amplitude

$$\theta_J(\mathcal{L}) = \left\| (1 - |\Psi\rangle\langle\Psi|)\rho_J(\mathcal{L})|\Psi\rangle \right\|_2, \tag{25}$$

which approximately describes the amount of coherent Hamiltonian errors (Extended Data Table 1). Here $\rho_J(\mathcal{L}) = (\mathcal{L} \otimes 1_{d^2})[|\Psi\rangle\langle\Psi|]$ is the Jamiołkowski state and $|\Psi\rangle$ is a maximally entangling four-qubit state that originates from the relation of quantum processes to states in a Hilbert space twice the dimension via the Choi–Jamiołkowski isomorphism[52]. For small errors, the average gate infidelity can be approximated by[30]

$$1 - F_{\text{gate}} = \frac{d}{d+1}\left[\epsilon_J(\mathcal{L}) + \theta_J(\mathcal{L})^2\right]. \tag{26}$$

For a comparison of the performance of the single-qubit gates with previous experiments reporting single-qubit gate fidelities, we compute the fidelities projected to the single-qubit space from the PTMs or the error generators. In Fig. 2 and Extended Data Fig. 2, single-qubit gate fidelities are estimated by projecting the PTMs onto the corresponding subspace. Let $\mathcal{P}_j$ be the projector on the subspace of qubit $j$, then the fidelity is given by

$$F_{\text{sub}} = \frac{\operatorname{Tr}(\mathcal{P}_j \mathcal{M}_{\text{exp}}^{-1}\mathcal{P}_j\mathcal{M}_{\text{ideal}}) + (d/2)}{(d/2)((d/2)+1)}. \tag{27}$$

Error bars for the fidelity projected to the subspace are computed using standard error propagation of the confidence intervals of $\mathcal{M}_{\text{exp}}$ provided by the pyGSTi package. A more optimistic estimation for the fidelities in the single-qubit subspace is given by projecting the error generators instead of the PTMs.

**Variational quantum eigensolver**
We follow the approach of ref. [36] to using the VQE algorithm to compute the ground-state energy of molecular hydrogen, after mapping this state onto the state of two qubits. We include this information here for completeness. The Hamiltonian of a molecular system in atomic units is

$$H = -\sum_i \frac{\nabla_{\mathbf{R}_i}^2}{2M_i} - \sum_j \frac{\nabla_{\mathbf{r}_j}^2}{2} - \sum_{i,j} \frac{Q_i}{|\mathbf{R}_i - \mathbf{r}_j|} + \sum_{i,j>i} \frac{Q_i Q_j}{|\mathbf{R}_i - \mathbf{R}_j|} + \sum_{i,j>i} \frac{1}{|\mathbf{r}_i - \mathbf{r}_j|}, \tag{28}$$

where $\mathbf{R}_i$, $M_i$ and $Q_i$ are the position, mass and charge, respectively, of the $i$th nuclei and $\mathbf{r}_j$ is the position of the $j$th electron. The first two sums describe the kinetic energies of the nuclei and electrons, respectively. The last three sums describe the Coulomb repulsion between nuclei and electrons, nuclei and nuclei, and electrons and electrons, respectively. As we are primarily interested in the electronic structure of the molecule, and nuclear masses are a few orders of magnitude larger than the electron masses, the nuclei are treated as static point charges

under the Born–Oppenheimer approximation. Consequently, the electronic Hamiltonian can be simplified to

$$H_e = -\sum_i \frac{\nabla_{\mathbf{r}_i}^2}{2} - \sum_{i,j} \frac{Q_i}{|\mathbf{R}_i - \mathbf{r}_j|} + \sum_{i,j>i} \frac{1}{|\mathbf{r}_i - \mathbf{r}_j|}. \tag{29}$$

Switching into the second-quantization representation, described by fermionic creation and annihilation operators, $a_p^\dagger$ and $a_q$, acting on a finite basis, the Hamiltonian becomes

$$H_e = \sum_{pq} h_{pq} a_p^\dagger a_q + \sum_{pqrs} h_{pqrs} a_p^\dagger a_q^\dagger a_r a_s, \tag{30}$$

where $p, q, r$ and $s$ label the corresponding basis states. The antisymmetry under exchange is retained through the anticommutation relation of the operators. The weights of the two sums are given by the integrals

$$h_{pq} = \int d\boldsymbol{\sigma}\,\psi_p^*(\boldsymbol{\sigma})\left(\frac{\nabla_{\mathbf{r}_i}^2}{2} - \sum_i \frac{Q_i}{|\mathbf{R}_i - \mathbf{r}|}\right)\psi_q(\boldsymbol{\sigma}), \tag{31}$$

$$h_{pqrs} = \int d\boldsymbol{\sigma}_1 d\boldsymbol{\sigma}_2 \frac{\psi_p^*(\boldsymbol{\sigma}_1)\psi_q^*(\boldsymbol{\sigma}_2)\psi_s(\boldsymbol{\sigma}_1)\psi_r(\boldsymbol{\sigma}_2)}{|\mathbf{r}_1 - \mathbf{r}_2|}, \tag{32}$$

where $\boldsymbol{\sigma}_i = (\mathbf{r}_i, s_i)$ is a multi-index describing the position $\mathbf{r}_i$ and the spin $s_i$ of electron $i$. Such a second-quantized molecular Hamiltonian can be mapped onto qubits using the Jordan–Wigner (JW) or the BK transformation[5]. The JW transformation directly encodes the occupation number (0 or 1) of the $i$th spin orbital into the state ($|0\rangle$ or $|1\rangle$) of the $i$th qubit. The number of qubits required after JW transformation is, thus, the same as the number of spin orbitals that are of interest. The BK transformation, on the other hand, encodes the information in both the occupation number and parities, whether there is an even or odd occupation in a subset of spin orbitals.

Taking molecular hydrogen in the HF basis as an example, we are interested in investigating the bonding ($|O_1\uparrow\rangle$, $|O_1\downarrow\rangle$) and the antibonding orbital state ($|O_2\uparrow\rangle$, $|O_2\downarrow\rangle$). The initial guess of the solution is the HF state in which both electrons occupy the $|O_1\rangle$ orbital. The JW transformation encodes the HF initial state as $|1100\rangle$, representing $|N_{O_1\downarrow} N_{O_1\uparrow} N_{O_2\downarrow} N_{O_2\uparrow}\rangle$ from left to right, where $N_{O_iS}$ is the occupation of the $O_iS$ spin orbital with $S = \uparrow, \downarrow$. The BK transformation encodes the HF initial state as $|1000\rangle$, where the first and the third qubits (counting from the right) encode the occupation number of the first and third spin orbitals ($N_{O_1\uparrow} = 1$ and $N_{O_2\uparrow} = 0$), the second qubit encodes the parity of the first two spin orbitals ($(N_{O_1\uparrow} + N_{O_1\downarrow})\bmod 2 = 0$) and the fourth qubit encodes the parity of all four spin orbitals ($(N_{O_1\uparrow} + N_{O_1\downarrow} + N_{O_2\uparrow} + N_{O_2\downarrow})\bmod 2 = 0$). With the standard transformation rules for fermionic creation and annihilation operators, the system Hamiltonian becomes a four-qubit Hamiltonian

$$\begin{aligned} H_{\text{JW}} = {}& g_0 I + g_1 Z_1 + g_2 Z_2 + g_3 Z_3 + g_4 Z_4 \\ & + g_5 Z_1 Z_2 + g_6 Z_1 Z_3 + g_7 Z_1 Z_4 \\ & + g_8 Z_2 Z_3 + g_9 Z_2 Z_4 + g_{10} Z_3 Z_4 \\ & + g_{11} Y_1 X_2 X_3 Y_4 + g_{12} Y_1 Y_2 X_3 X_4 \\ & + g_{13} X_1 X_2 Y_3 Y_4 + g_{14} X_1 Y_2 Y_3 X_4, \end{aligned} \tag{33}$$

$$\begin{aligned} H_{\text{BK}} = {}& g_0 I + g_1 Z_1 + g_2 Z_2 + g_3 Z_3 \\ & + g_4 Z_1 Z_2 + g_5 Z_1 Z_3 + g_6 Z_2 Z_4 \\ & + g_7 Z_1 Z_2 Z_3 + g_8 Z_1 Z_3 Z_4 + g_9 Z_2 Z_3 Z_4 \\ & + g_{10} Z_1 Z_2 Z_3 Z_4 + g_{11} X_1 Z_2 X_3 \\ & + g_{12} Y_1 Z_2 Y_3 + g_{13} X_1 Z_2 X_3 Z_4 + g_{14} Y_1 Z_2 Y_3 Z_4. \end{aligned} \tag{34}$$

The subscripts are used to label the qubits. We see that, owing to the symmetry of the represented system in $H_{BK}$, qubit 2 and qubit 4 are never flipped, allowing us to reduce the dimension of the Hamiltonian to

$$
\begin{aligned}
H_{BK}^{reduced} = \; & h_0 I + h_1 Z_1 + h_2 Z_2 + h_3 Z_1 Z_2 \\
& + h_4 X_1 X_2 + h_5 Y_1 Y_2 h_0 \\
& + h_1 ZI + h_2 IZ + h_3 ZZ \\
& + h_4 XX + h_5 YY,
\end{aligned}
\tag{35}
$$

where qubit 1 has been relabelled as qubit 2 and qubit 3 has been relabelled as qubit 1. The HF initial state is, therefore, reduced to $|01\rangle$ and the Hamiltonian is rephrased to be consistent with the partial tomography expression in Fig. 5. This reduced representation requires only two qubits to simulate the hydrogen molecule. We emphasize that such a reduction of the BK Hamiltonian is not a special case for the $H_2$ molecule but is connected to symmetry considerations to reduce the complexity of systems, in a scalable way.

VQE is a method to compute the ground-state energy of the Hamiltonian. The total energy can be directly calculated by measuring the expectation value of each Hamiltonian term. This can be done easily by partial quantum state tomography. All the expectation values are then added up with a set of weights ($h_0$ through $h_5$). The weights are only functions of the internuclear separation ($R$) and can be computed efficiently by a classical computer. Here we use the OpenFermion Python package to compute these weights[38].

The main task of the quantum processor is, then, to encode the molecular spin-orbital state into the qubits. The starting point is the HF initial state, which is believed to largely overlap with the actual ground state. In order to find the actual ground state, the initial state needs to be 'parameterized' into an ansatz to explore a subspace of all possible states. We apply the unitary coupled cluster (UCC) theory to the parameterized ansatz state, which is used to describe many-body systems and cannot be efficiently executed on a classical computer[53]. The UCC operator has a format

$$
U_{UCC}(\boldsymbol{\theta}) = e^{\sum_n (T_n(\boldsymbol{\theta}) - T_n^\dagger(\boldsymbol{\theta}))},
\tag{36}
$$

with

$$
T_1(\boldsymbol{\theta}) = \sum_{m,i} \boldsymbol{\theta}_i^m a_m^\dagger a_i,
\tag{37}
$$

$$
T_2(\boldsymbol{\theta}) = \sum_{m,n,i,j} \boldsymbol{\theta}_{i,j}^{m,n} a_m^\dagger a_n^\dagger a_i a_j
\tag{38}
$$

representing single and double excitation of the electrons. The indices $i$ and $j$ label the occupied spin orbitals and $m$ and $n$ are the labels of the unoccupied spin orbitals. The vector $\boldsymbol{\theta}$ is the set of all parameters to optimize. In the case of a $H_2$ molecule, the UCC operator is transformed into a qubit operator as

$$
U_{UCC}^{BK}(\boldsymbol{\theta}) = e^{-i\theta XY},
$$

where $\theta$ is a single parameter to variationally optimize.

## Data availability

Data supporting this work are available at Zenodo, https://doi.org/10.5281/zenodo.5044450.

## Code availability

The codes used for data acquisition and processing are from the open source Python packages QCoDeS (https://github.com/QCoDeS/Qcodes), QTT (https://github.com/QuTech-Delft/qtt) and PycQED (https://github.com/DiCarloLab-Delft/PycQED_py3). The codes used for the design and analysis of the gate-set tomography experiment are from pyGSTi (https://github.com/pyGSTio/pyGSTi). The codes used for the design and analysis of the variational quantum eigensolver experiment are from OpenFermion (https://github.com/quantumlib/OpenFermion).

43. Reed, M. *Entanglement and Quantum Error Correction with Superconducting Qubits*. PhD thesis, Yale Univ. (2013).
44. Russ, M. et al. High-fidelity quantum gates in Si/SiGe double quantum dots. *Phys. Rev. B* **97**, 085421 (2018).
45. Yang, Y.-C., Coppersmith, S. N. & Friesen, M. Achieving high-fidelity single-qubit gates in a strongly driven charge qubit with 1/*f* charge noise. *npj Quantum Inf.* **5**, 12 (2019).
46. Koski, J. V. et al. Strong photon coupling to the quadrupole moment of an electron in a solid-state qubit. *Nat. Phys.* **16**, 642–646 (2020).
47. Russ, M., Philips, S., Xue, X. & Vandersypen, L. M. K. The path to high fidelity multi-qubit gates for quantum dot spin qubits. *Bull. Am. Phys. Soc.* **66**, abstr. S29.00002 (2021).
48. Greenbaum, D. Introduction to quantum gate set tomography. Preprint at https://arxiv.org/abs/1509.02921 (2015).
49. Nielsen, E., Blume-Kohout, R. J., Rudinger, K. M., Proctor, T. J., Saldyt, L. & USDOE, *Python GST Implementation (PyGSTi) v. 0.9*, Tech. Rep. PyGSTi (Sandia National Laboratories, 2019).
50. Nielsen, E. et al. Probing quantum processor performance with pyGSTi. *Quantum Sci. Technol.* **5**, 044002 (2020).
51. White, A. G. et al. Measuring two-qubit gates. *J. Opt. Soc. Am. B* **24**, 172–183 (2007).
52. Jamiolkowski, A. Linear transformations which preserve trace and positive semidefiniteness of operators. *Rep. Math. Phys.* **3**, 275–278 (1972).
53. Taube, A. G. & Bartlett, R. J. New perspectives on unitary coupled-cluster theory. *Int. J. Quantum Chem.* **106**, 3393–3401 (2006).

**Acknowledgements** We acknowledge discussions with P. Cerfontaine, C. Bureau-Oxton, M. T. Madzik, A. Morello, J. Helsen, B. Terhal, M. Veldhorst and all the members of the spin qubit team, and technical assistance from O. Benningshof, M. Sarsby, R. Schouten and R. Vermeulen. This research was funded by the Dutch Ministry of Economic Affairs through the allowance for Top Consortia for Knowledge and Innovation (TKI) and the Army Research Office (ARO) under grant number W911NF-17-1-0274. The views and conclusions contained in this document are those of the authors and should not be interpreted as representing the official policies, either expressed or implied, of the ARO or the US Government. The US Government is authorized to reproduce and distribute reprints for government purposes notwithstanding any copyright notation herein.

**Author contributions** X.X. performed the experiment, with help from N.S. and B.U. M.R. developed the theory model and analysed the data with X.X. N.S. fabricated the quantum dot device. A.S. and G.S. designed and grew the Si/SiGe heterostructure. X.X. and L.M.K.V. conceived the project. L.M.K.V. supervised the project. X.X., M.R. and L.M.K.V. wrote the manuscript, with input from all authors.

**Competing interests** The authors declare no competing interests.

**Additional information**
**Correspondence and requests for materials** should be addressed to Lieven M. K. Vandersypen.

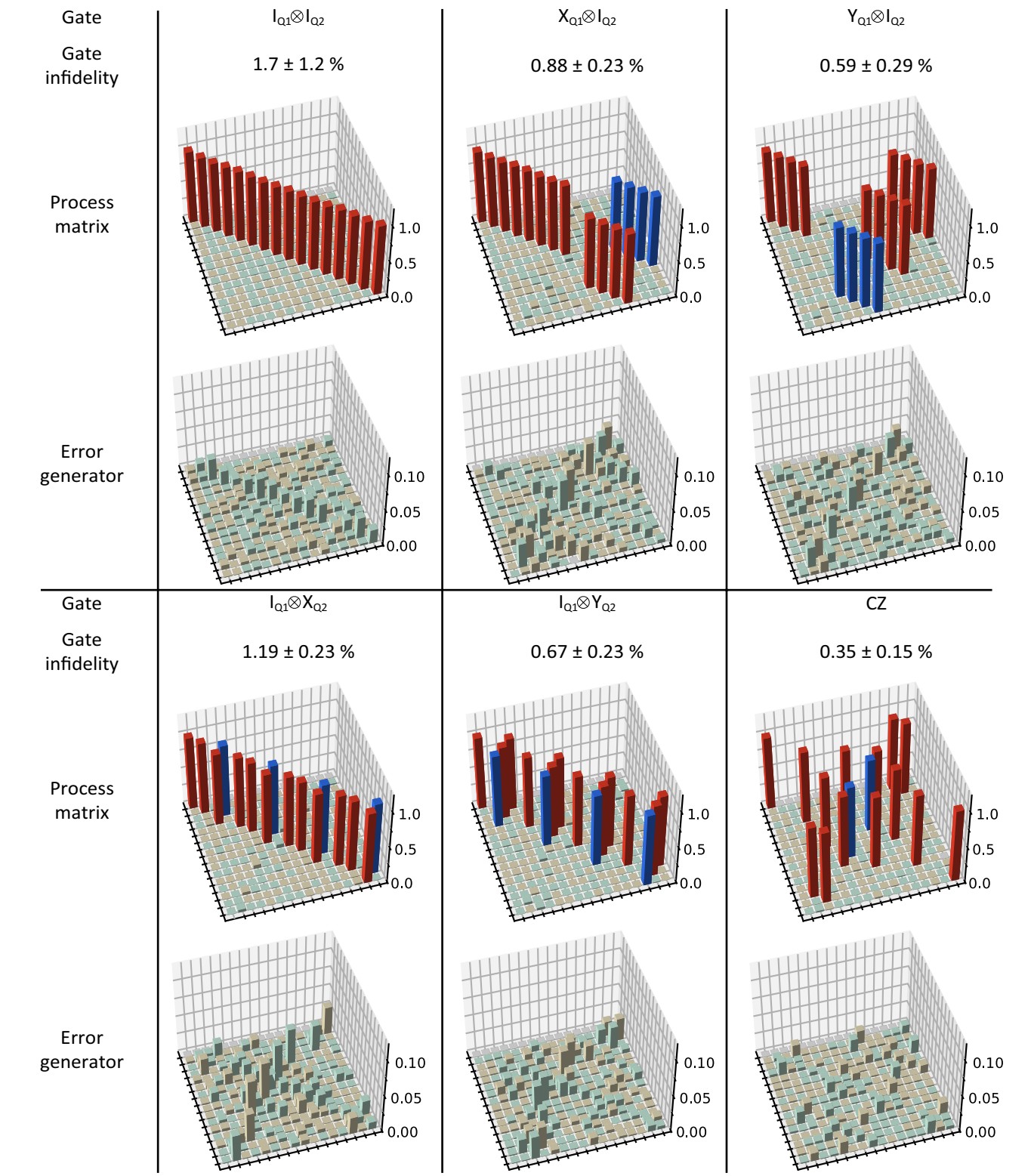

**Extended Data Fig. 1 | Two-qubit processes.** Average gate infidelities, process matrices (PTMs) and error generators of the six quantum gates in the chosen gate set. These results are analysed by the pyGSTi package using maximum-likelihood estimation.

| Gate | $I_{Q1}$ | $X_{Q1}$ | $Y_{Q1}$ |
|---|---|---|---|
| Gate infidelity | 0.75 ± 0.33 % | 0.320 ± 0.073 % | 0.27 ± 0.57 % |
| Process matrix | | | |

| Gate | $I_{Q2}$ | $X_{Q2}$ | $Y_{Q2}$ |
|---|---|---|---|
| Gate infidelity | 1.11 ± 0.39 % | 0.39 ± 0.68 % | 0.131 ± 0.025 % |
| Process matrix | | | |

**Extended Data Fig. 2 | Single-qubit processes.** Average gate infidelities and process matrices (PTMs) of the identity gates (idle gates) and single-qubit X/Y gates in the subspace of the individual qubits. The individual PTMs are calculated from the PTMs in the two-qubit space (see Methods).

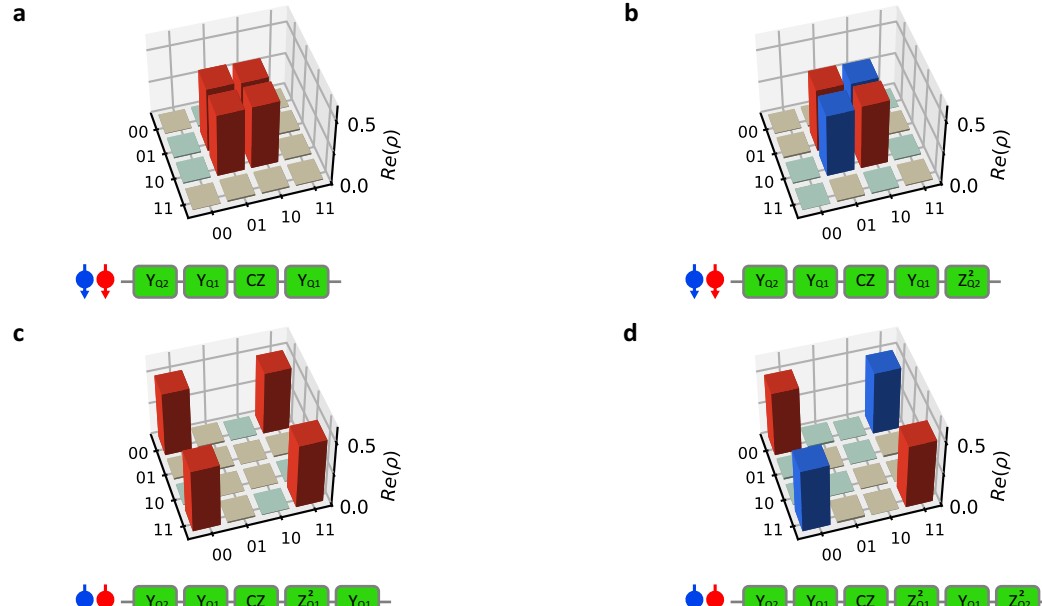

**Extended Data Fig. 3 | Bell states predicted from the quantum processes.** Top panels show the real part of the reconstructed density matrices of the four Bell states $|\Psi^+\rangle = (|01\rangle + |10\rangle)/\sqrt{2}$ (**a**), $|\Psi^-\rangle = (|01\rangle - |10\rangle)/\sqrt{2}$ (**b**), $|\Phi^+\rangle = (|00\rangle + |11\rangle)/\sqrt{2}$ (**c**) and $|\Phi^-\rangle = (|00\rangle - |11\rangle)/\sqrt{2}$ (**d**). The colour code is the same as in Fig. 4. Bottom panels show the quantum circuit used to reconstruct the Bell states. $Z^2_{Qi}$ is a virtual π-rotation around the $\hat{z}$ axis on the $i$th qubit, which is executed by a phase update on the microwave reference clock of the qubit and, therefore, is error-free. We numerically estimate the state fidelities to be 98.42% for the $|\Psi^+\rangle$ and $|\Psi^-\rangle$ states and 97.75% for the $|\Phi^+\rangle$ and $|\Phi^-\rangle$ states.

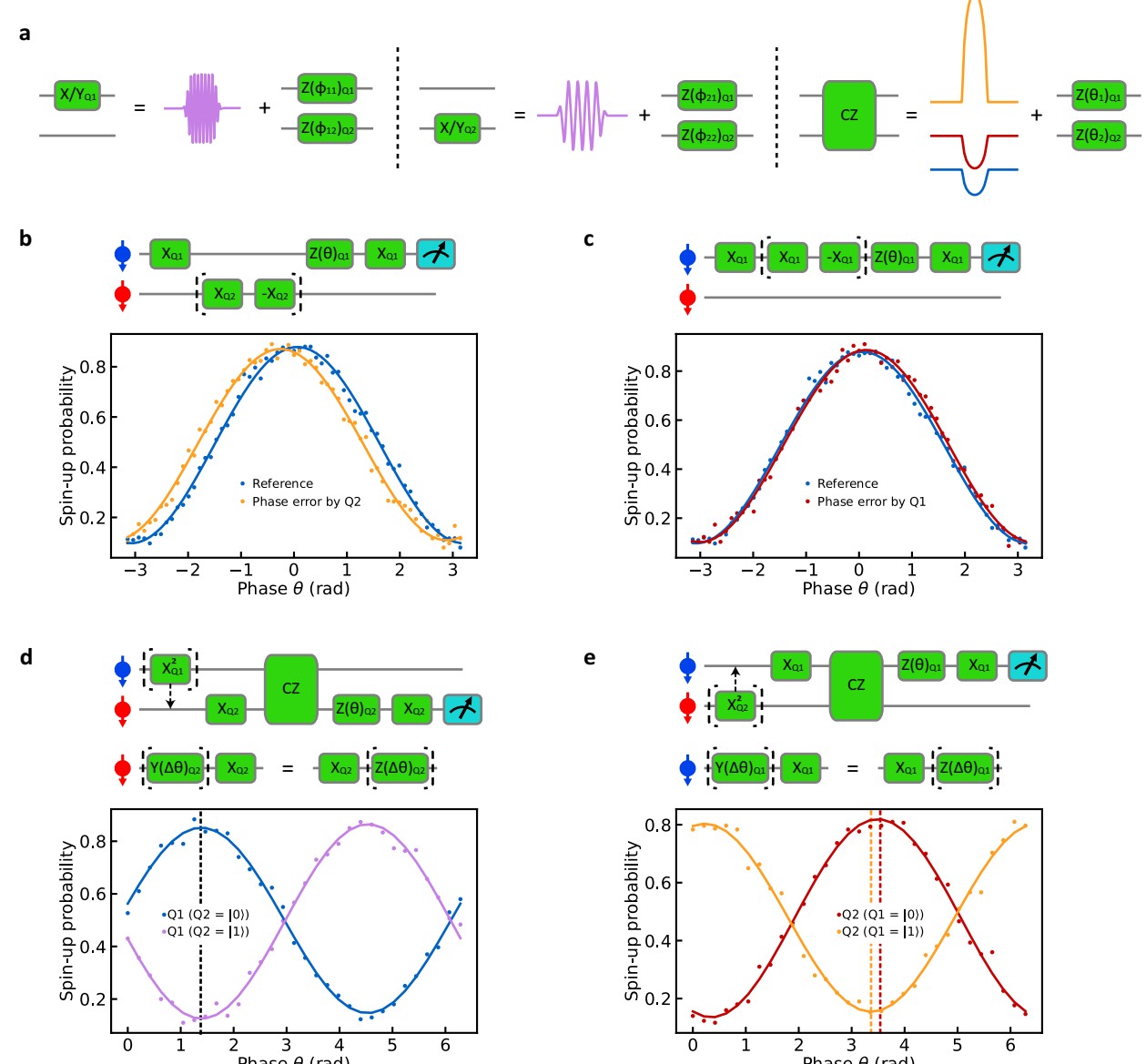

**Extended Data Fig. 4 | Initial gate calibrations. a**, Decomposition of single-qubit and two-qubit gates. After each microwave burst for single-qubit rotations, a corresponding phase correction is applied to each qubit. The CZ gate is implemented by a barrier voltage pulse applied to gate B (orange) and negative compensation pulses applied to gates LP (blue) and RP (red), with the same shape as the barrier pulse. Single-qubit phase corrections are then applied on each qubit to compensate the frequency detuning induced by electron movement in the magnetic field gradient. **b**, **c**, Calibration of phase corrections on $Q_1$ induced by a single-qubit gate applied on $Q_2$ ($\phi_{21}$, **b**) and on $Q_1$ ($\phi_{11}$, **c**). A relative phase shift, $2\phi_{21}$ ($2\phi_{11}$), is determined by interleaving the target gate (a $\pi/2$ rotation) and its inverse (a $-\pi/2$ rotation) on $Q_2$ ($Q_1$) in a

Ramsey interference sequence. **d**, **e**, Calibration of phase corrections on each qubit after the CZ gate, using $Q_1$ (**d**) and $Q_2$ (**e**) as the control qubits, respectively. When the amplitude of the barrier pulse is perfectly calibrated, the two curves in each experiment should be out of phase by 180°. However, when the barrier pulse amplitude is calibrated such that one of the two experiments shows a 180° phase difference (**d**), the phase difference in the other calibration experiment always deviates by a few degrees. One possible explanation is that the optional $\pi$-rotation applied to the control qubit induces a small, off-resonance rotation on the other qubit, causing an additional phase on the target qubit to appear in the measurement due to the commutation relation of the Pauli operators.

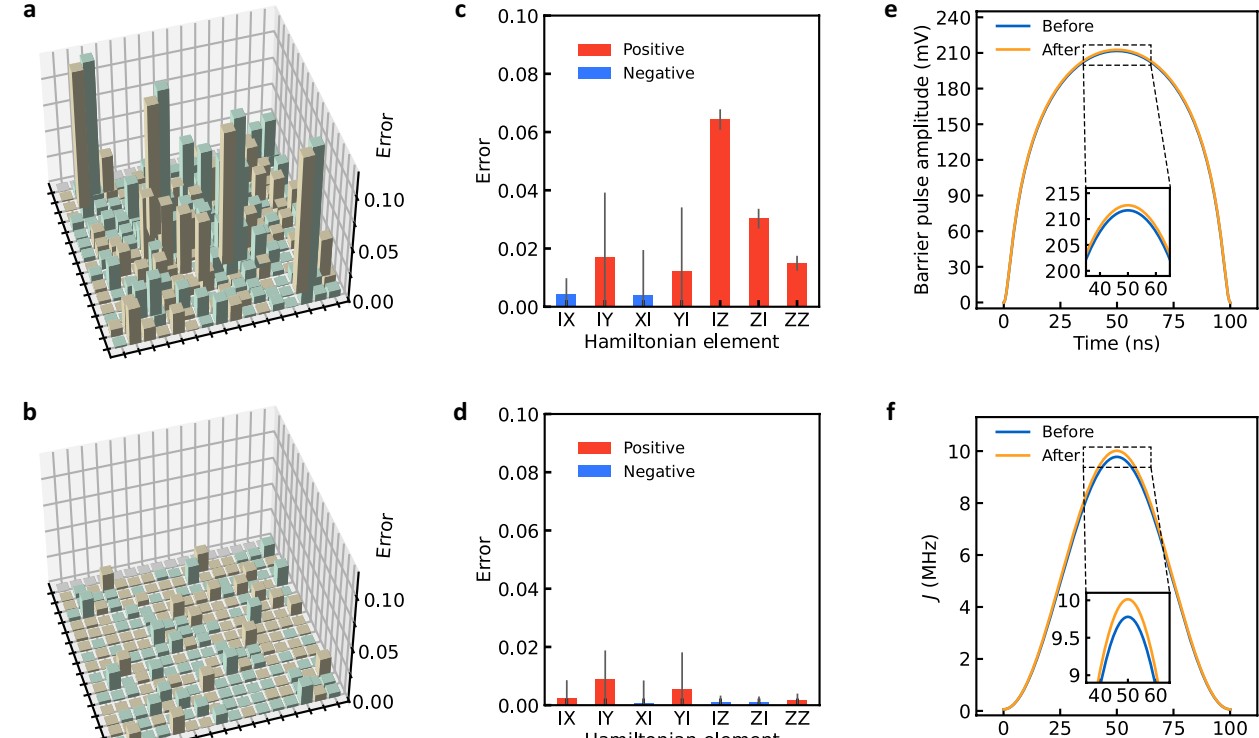

**Extended Data Fig. 5 | Pulse optimization. a**, **b**, Full error generators for a CZ gate calibrated by conventional Ramsey sequences (**a**) and after improving the calibration using the information extracted from **a** (**b**), resulting in fidelities of 97.86% and 99.65%, respectively. **c**, **d**, Seven Hamiltonian errors (IX, IY, XI, YI, IZ, ZI and ZZ) extracted from the error generators shown in **a** (**c**) and **b** (**d**). Owing to the crosstalk-induced additional phases shown in Extended Data Fig. 4, errors IZ, ZI and ZZ occur systematically in conventional calibrations. Error bars indicate the 2$\sigma$ confidence intervals computed using the Hessian of the loglikelihood function. **e**, **f**, Shapes of the barrier pulses (**e**) and their

corresponding $J$ envelopes (**f**) for a CZ gate before and after being corrected by GST. Since the Hamiltonian to generate a CZ gate is $H = (II + IZ + ZI - ZZ)/2$, the positive ZZ error shown in **c** is corrected by increasing the amplitude of the pulse. The IZ and ZI errors are corrected by decreasing the phase shifts $\theta_1$ and $\theta_2$ after the CZ gate. Hamiltonian errors in single-qubit gates are corrected similarly. The results presented in **b** and **d** are achieved in four loops of correction, with each loop correcting the parameters by approximately 70% of the measured deviation.

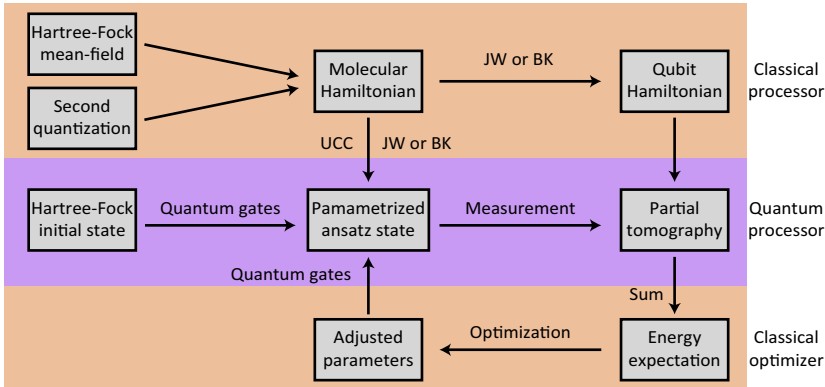

**Extended Data Fig. 6 | Workflow of the variational quantum eigensolver algorithm.** The qubit Hamiltonian is typically transformed from the molecular Hamiltonian by JW transformation or BK transformation by a classical processor (see Methods). A HF initial state is encoded into the qubit states according to JW or BK transformation and then transformed by the quantum processor into a parameterized ansatz state by considering single and double excitation in the molecule using the UCC theory. The expectation value of each individual Hamiltonian term is directly measured by partial state tomography. The expectation of the total energy is then calculated by the weighted sum of the individual expectations. The result is fed into a classical optimizer, which suggests a new parameterized ansatz state for the next run. This process is repeated until the expectation of the total energy converges.

**Extended Data Table 1 | Gate metrics**

| | $1 - F_{\text{gate}}$ | $1 - F_{\text{sub}}$ | $\epsilon_J$ | $\theta_J$ | $D$ | $\|\cdot\|_\diamond$ |
|---|---|---|---|---|---|---|
| $I$ | $0.017 \pm 0.012$ | Q1: $0.0075 \pm 0.0033$<br>Q2: $0.0111 \pm 0.0039$ | $0.021$ | $0.0097$ | $0.024 \pm 0.015$ | $0.038 \pm 0.019$ |
| $X_{\text{Q1}}$ | $0.0088 \pm 0.0023$ | $0.00320 \pm 0.00073$ | $0.010$ | $0.027$ | $0.032 \pm 0.022$ | $0.047 \pm 0.035$ |
| $Y_{\text{Q1}}$ | $0.0059 \pm 0.0029$ | $0.0027 \pm 0.0057$ | $0.0069$ | $0.022$ | $0.0256 \pm 0.0073$ | $0.034 \pm 0.022$ |
| $X_{\text{Q2}}$ | $0.0119 \pm 0.0023$ | $0.0039 \pm 0.0068$ | $0.014$ | $0.028$ | $0.035 \pm 0.030$ | $0.044 \pm 0.041$ |
| $Y_{\text{Q2}}$ | $0.0067 \pm 0.0023$ | $0.00131 \pm 0.00025$ | $0.0079$ | $0.022$ | $0.0265 \pm 0.0080$ | $0.034 \pm 0.014$ |
| CZ | $0.0035 \pm 0.0015$ | — | $0.0042$ | $0.016$ | $0.018 \pm 0.014$ | $0.023 \pm 0.010$ |

Detailed overview of important metrics of the gate set {I, $X_{\text{Q1}}$, $Y_{\text{Q1}}$, $X_{\text{Q2}}$, $Y_{\text{Q2}}$, CZ}: the average gate fidelity $F_{\text{gate}}$ (see equation (22)) and the fidelity reduced to the single-qubit subspace $F_{\text{sub}}$ (see equation (27)), the Jamiołkowski probability $\epsilon_J$ (see equation (24)), the Jamiołkowski amplitude $\theta_J$ (see equation (25)), the trace distance $D(\mathcal{M}_{\text{ideal}}, \mathcal{M}_{\text{exp}}) = \|\mathcal{M}_{\text{ideal}} - \mathcal{M}_{\text{exp}}\|_1/2$ and the diamond norm $\|\mathcal{M}_{\text{ideal}}, \mathcal{M}_{\text{exp}}\|_\diamond = \max_\rho \|(\mathcal{M}_{\text{ideal}} \otimes 1_{d^2})\rho - (\mathcal{M}_{\text{exp}} \otimes 1_{d^2})\rho\|_1/2$.