## [Peer Review File · Nature]

Manuscript Title: Quantum logic with spin qubits crossing the surface code threshold

Reviewer Comments & Author Rebuttals

Reviewer Reports on the Initial Version:

Referee #1 (Remarks to the Author):

In this work the authors realized a high-fidelity two-qubit (CPHASE) gate for semiconductor spin qubits. Two-qubit gate fidelity is an important metric, and the reported value of 99.65% presents a considerable improvement from the previous record in this system (roughly speaking, a reduction of error by a factor of 5). Recent demonstrations of >99.5% exchange-based gates between two spins in the singlet-triplet subspace [Cerfontaine, P. et al. Nature Communications 11:4144 (2020); Takeda, K. et al. Physical Review Letters 124, 117701 (2020)] have lead to optimism that two-qubit gate fidelities can be equally high once they are fully calibrated, but it has remained a nontrivial task experimentally. This paper seems to present a concrete calibration procedure and demonstrate that it is indeed practically possible.

The main claim of the manuscript is that they can now compute at the surface code error threshold with semiconductor spin qubits, making this work potentially publishable in Nature. However, I am concerned that this main claim is not warranted by the argument given in the current manuscript. Computing at the surface code error threshold means that the logical error rate does not grow as the size of the qubit array is increased, rather than individual errors being below ~1%. This is because these thresholds are calculated for other qubit platforms and assuming significantly different constraints. The authors need to show that with the demonstrated errors in single- and two-qubit gates, such a fault-tolerant behavior will be realized, after explicitly stating the required assumptions on other parameters such as SPAM errors, QND measurement time etc. (as was done in e.g. Barends, R. et al. Nature 508, 500 (2014)).

The authors employed a VQE algorithm to showcase the performance of their device. However, the implication of their result does not seem to be clearly presented. The authors have to acknowledge the fact that this is not the first quantum algorithm run on silicon quantum processors [Ref. 17 from the same group]. The achieved accuracy (20 milliHartree) is still roughly an order of magnitude above the chemical accuracy threshold - the level required to make useful predictions. Given these, I believe the authors need to elaborate on the significance of the result and the future prospects in this direction.

In addition, the current manuscript does not seem to present a compelling validation of the custom fidelity analysis designed in this study. The authors used gate set tomography (GST) to characterize gate fidelities, which is established in the standard form; however, the one employed in this work is customized by the authors. Given that self-consistent tomographic reconstruction is a highly nonlinear and nontrivial estimation process, shouldn't one naively expect the reliability to be influenced by choice of the gate set and model? Despite this, only a crude indicator of goodness of fit (four out of five stars) is given. The validation of their GST design and analysis would therefore deserve a more detailed discussion. Instead, a consistent measurement of a CPHASE gate fidelity by randomized benchmarking will help support the claimed fidelity in a more accessible manner.

To conclude, given these issues above, I believe that this manuscript does not convincingly present the level of an advance that Nature readers will find as compelling as some of the other papers published by these same authors in this journal. I imagine that after these issues are addressed, the paper will be very well suited to a slightly more specialized journal.

Below I list some minor comments and suggestions.

(1) Does any observation (e.g. in Fig. 1(d)) indicate that avoiding a tilt is a good strategy for high-fidelity exchange-based gates in this device too, or is it assumed to be optimal based on previous studies?

(2) The single qubit gate fidelity in the full two-qubit space (99.16% on average) is reduced from the one in the single-qubit subspace. What is the reduction when only idling infidelities of $\sim 1\%$ are considered? Do other mechanisms (such as crosstalk) contribute significantly at this stage?

(3) Can the authors consider reporting Bell state fidelities uncorrected for SPAM errors as well (in Fig. 4c and Extended Data Fig. 3)?

(4) On line 72 of page 6 - what do the authors mean by the linear response regime?

(5) I believe there are typos in the domains of t given in Eq. (1).

(6) On line 36 of page 7, can the authors specify how many GST loops were (typically) required in this study?

(7) The order of terms in Eq. (2) does not match the explanation given in the same paragraph. I also believe that \hbar in the definition of \vec{S} should be dropped.

(8) On line 69 of page 7 - I suppose the word "amounts" should read "accounts".

(9) On line 78 of page 7 - I suppose the word "loose" should read "lose".

(10) On line 88 of page 7 - I suppose δf_{Q1} should be δf_{Q2} .

(11) On line 10 of page 8, the authors obtained $\gamma \sim 1.2$. Is this fully consistent with the statement on line 4 of page 4?

(12) There's a typo in Eq. (12).

(13) In Eq. (18), information about the value of frequency should be included, using a standard notation used on line 33 of page 8.

(14) Can the authors clarify the difference (in the meaning) between a predicted gate infidelity of 0.022% and an estimated fidelity $> 99.97\%$ on lines 29 and 37 on page 9.

(15) On line 94 of page 9, there appears to be something wrong with the equation relating F_{ent} and F_{gate} .

(16) I encourage the authors to check Methods section VIII again. Many parameters seem undefined (for instance, p, q, r, s in Eq. (30) and σ, r_i and i in Eq. (31)) and I believe there are some typos (at least in Eqs. (33), (34) and (35) and possibly in Eq. (31)). Furthermore, I am confused from which side the qubit index is counted after the authors reduced the dimension using the BK transformation and implicitly relabeled the qubits. (It should be consistent with the main text, e.g. Fig. 5b for readability.)

(17) Extended Data Fig. 3 - there's a typo in the Bell state in the figure caption for panel (c). Can the authors briefly explain why the estimated state fidelities are the same for $|\Psi^{+-}\rangle$ and $|\Psi^{-+}\rangle$ ($|\Phi^{+-}\rangle$ and $|\Phi^{-+}\rangle$) - i.e. any reasons why the estimated fidelity of virtual Z gate is 1?

(18) Extended Data Fig. 5 - can the authors give error bars in panels c and d and explain (preferably in the main text) how they derived the error bars for the reported value of CPHASE fidelity?

Referee #2 (Remarks to the Author):

In the manuscript "Computing with spin qubits at the surface code error threshold," the authors report a spin-based two-qubit quantum processor with average single- and two-qubit gate fidelities above 99%, reaching for the first time the generic error threshold for the surface code in this system. Using standard Ramsey interference experiments and gate set tomography, the authors provide detailed characterizations of device properties as well as the full Pauli transfer matrices and error generators describing the primitive quantum gate set used. They then go on to demonstrate the capabilities of their device by running a variational quantum eigensolver algorithm and computing the dissociation energy for molecular hydrogen.

The achievement of a spin-based quantum processor with two-qubit gate fidelities above 99% represents a major milestone for solid-state semiconductor quantum computing platforms. Additionally, the manuscript is detailed, clearly written, and well-organized. I also find the detailed characterization of the device and the various noise sources helpful for the community to perform follow-up theoretical work on this platform. For these reasons, it meets the high standards of Nature provided the following points are addressed:

1. Some aspects may be hard to follow for the spin qubit community. While verification and validation are critical aspects of building a quantum processor, not everyone is familiar with some of the relevant concepts. More specifically, the readers would benefit by understanding better the concept of germ sequences and how these are chosen.
2. The text refers to Figure 1e, however, there is not (e) panel in Fig. 1.
3. I find the last sentence of the abstract a bit odd and would suggest to the authors to rephrase it. In particular, the comment that now spin qubits 'have gained credibility' is not a scientific statement, and it's something that the authors cannot claim.
4. It is interesting and helpful that the authors characterize single-qubit gate fidelities in the two-qubit system. As they say, this fidelity is distinct (lower) from single-qubit gate fidelities in a single-qubit system. For the same reason, the two-qubit gate fidelity will be lowered in a three-qubit (or larger) system. It would be helpful to have a relevant comment about this point and its repercussions for scalability. Given what the present experiment has shown, can the authors speculate about the expected two-qubit fidelity in a larger system?
5. Typically, the CPHASE gate with the phase = π is called CZ. It is probably clearer to use this more standard term.
6. The theoretical Bell state fidelity is reported, but the experimental number is not given. Why is that?
7. While overlapping gates in the QD device have been shown by other groups to lower the noise, the authors don't seem to have adopted it. Is there a reason for that?
8. In the Methods section, it is mentioned that the Ramsey sequences "are automatically executed every two hours". This statement is unclear to me—this seems like a long timescale, where does it come from?
9. Typo in Sec. III: "loose information" should be "lose information".

10. In Sec. VII some gate durations are given, but others are not.

11. When UCC is introduced, the authors write that it "is widely believed to be a powerful approach". I would say that UCC is a standard approach used in experimental proof of principle demonstrations, but it is far from state-of-the-art in the theoretical VQE community.

Referee #3 (Remarks to the Author):

In this manuscript the authors experimentally probe the performance of a 2 qubit quantum information processor implemented in a silicon quantum dot double well. The qubits are defined as single electron spin states, one per well. Single qubit gates are performed through EDSR and the two qubit interaction comes from electro-statically modulating the tunnel barrier in order to induce an exchange interaction between the two spins. The authors use gate set tomography to measure two qubit gate fidelities greater than 99% and proceed to run a small variational quantum algorithm.

There have been a couple demonstrations of 2-qubit gates in semiconductor systems, but as far as this referee is aware this is currently the highest fidelity in dots by a fair margin and probably the first demonstration that puts semiconductor qubits in contention with superconductor or ion quantum computing implementations. For that reason I recommend this for publication in Nature with a couple minor revisions.

I suppose I have some bias that spin qubits are broadly interesting, and that an experiment at the forefront of spin qubit implementations should be a shoe-in for Nature. With that perspective, my general comments to the authors to improve this manuscript would be an expanded discussion about what challenges remain in order to make this a truly competitive quantum computation platform.

1. Gageset tomography is a fine method for extracting gate error but I find it odd that the authors haven't shown any results from randomized benchmarking (RB). For some time now RB has basically been the industry standard for reporting the performance of one and two qubit devices and omitting any results seems like an oversight.

I find this omission troubling because I can imagine several reasons why it might be hard to run RB on a dot system. DC electronics can be tricky, and in order to quantify errors at the $1e-3$ level (which probably amounts to 1000's of Clifford gate) I could imagine issues with stability or waveform memory. If long sequences of gates are hard for some reason I still absolutely believe this result should be in Nature, but I think it would behoove the authors to point out the open research and engineering problems in this promising platform. If there was no reason not to do RB in the first place, then do it, and the results will be all the stronger.

2. When I first saw the title of this work I thought I'd been handed a paper out of a time-capsule from the superconductor community, where there were manuscripts like "Superconducting quantum circuits at the surface code threshold for fault tolerance" out of the UCSB (later Google) group and "Universal Quantum Gate Set Approaching Fault-Tolerant Thresholds with Superconducting Qubits" from IBM. If this was intended to be a cute reference to those works it's fine, but I think there are some lessons to be drawn from them as well.

So, short of these titles being a touch hyperbolic, the main issue is that neither IBM nor Google have demonstrated fault-tolerance despite being at the "threshold" 7-8 years ago. The issue, of course, has been scaling. There was a promise in superconducting qubits, similar to those made by some semiconductor groups, that scaling is somehow a straightforward engineering problem once

one has demonstrated high fidelity one and two-qubit gates. It is a little hard to look at the pictures of the device in fig 1 and immediately see how this technology is scalable. Perhaps some discussion of the challenges moving forward in terms of engineering and fabrication would be interesting to the broader audience.

Referee nr. 1 (Remarks to the Author):
In this work the authors realized a high-fidelity two-
qubit (CPHASE) gate for semiconductor spin qubits.
Two-qubit gate fidelity is an important metric, and
the reported value of 99.65% presents a considerable
improvement from the previous record in this system
(roughly speaking, a reduction of error by a factor of
5). Recent demonstrations of >99.5% exchange-based
gates between two spins in the singlet-triplet subspace
[Cerfontaine, P. et al. Nature Communications 11:4144
(2020); Takeda, K. et al. Physical Review Letters 124,
117701 (2020)] have lead to optimism that two-qubit gate
fidelities can be equally high once they are fully cali-
brated, but it has remained a nontrivial task experimen-
tally. This paper seems to present a concrete calibration
procedure and demonstrate that it is indeed practically
possible.
We thank the referee for reviewing the manuscript and
for the helpful comments. We give our response below
and have modified our manuscript accordingly.
The main claim of the manuscript is that they can
now compute at the surface code error threshold with
semiconductor spin qubits, making this work potentially
publishable in Nature. However, I am concerned that this
main claim is not warranted by the argument given in the
current manuscript. Computing at the surface code error
threshold means that the logical error rate does not grow
as the size of the qubit array is increased, rather than
individual errors being below 1%. This is because these
thresholds are calculated for other qubit platforms and
assuming significantly different constraints. The authors
need to show that with the demonstrated errors in single-
and two-qubit gates, such a fault-tolerant behavior will
be realized, after explicitly stating the required assump-
tions on other parameters such as SPAM errors, QND
measurement time etc. (as was done in e.g. Barends, R.
et al. Nature 508, 500 (2014)).
We thank the referee for the critical comment. We
point out that our main claim is that the error rates of
the native gate set achieve the most widely used thresh-
old value required to achieve fault-tolerance and that
there is no limitation at the fundamental physics level
for spin qubits to reach such high gate fidelities. This
is clearly different from claiming a system to be fault-
tolerant, which no existing paper can claim. To make this
more explicit, we have reworded the last sentence of the
abstract as follows: “Having surpassed the 99% barrier
for the two-qubit gate fidelity, semiconductor qubits are
well positioned on the path to fault-tolerance and to possi-
ble applications in the era of noisy intermediate-scale
quantum (NISQ) devices.” Similarly, we have adapted
the concluding paragraph as follows: “Combining high-
fidelity initialization, readout, and control into a demon-
stration of fault-tolerance poses several key challenges to
be overcome. First, sufficiently large and reliable quan-
tum dot arrays must be constructed, with good connec-
tivity between the qubits. Second, the fidelities achieved
in small-scale systems must be maintained across such
larger systems, which will require reducing idling and
crosstalk errors”, in order to highlight the outstanding
challenges in achieving fault-tolerance.
We could in principle perform a similar simulation as
Barends et al. However, we also realize that such a sim-
ulation would need to rely on many assumptions on how
the gate errors, SPAM errors, and QND measurement
time will change (or not) as we extrapolate to a larger
system. In the literature, such extrapolations have been
made for spin qubits (D. K. Tuckett et al., PRL (2018);
D. K. Tuckett, et al., PRX (2019); D. K. Tuckett et al.,
PRL (2020)), and the threshold values are actually more
favorable than those of the surface code under depolar-
izing noise, taking advantage of the very long T_1 of spin
qubits. The 1% error threshold of the surface code under
depolarizing noise still serves as a widely used bench-
mark, which we adopt here.
The authors employed a VQE algorithm to showcase
the performance of their device. However, the implica-
tion of their result does not seem to be clearly presented.
The authors have to acknowledge the fact that this is not
the first quantum algorithm run on silicon quantum pro-
cessors [Ref. 17 from the same group]. The achieved
accuracy (20 milliHartree) is still roughly an order of
magnitude above the chemical accuracy threshold - the
level required to make useful predictions. Given these, I
believe the authors need to elaborate on the significance
of the result and the future prospects in this direction.
This is indeed not the first quantum algorithm im-
plemented with spin qubits. However, different from
the textbook algorithms such as Grover algorithm and
Deutsch-Jozsa algorithm, the VQE algorithm can be used
in real-world applications of future quantum computers.
Indeed VQE experiments have been used as a very pow-
erful quantitative benchmark for error mitigation (Kan-
dala et al., Nature (2017); Kandala et al., Nature (2019)).
Therefore, we consider this experiment as an important
reference for the spin qubit community to work on er-
ror mitigation of quantum gates, designing readout cir-
cuits, and even improving characterization and verifica-
tion methods. In order to make the motivation more
clear, we have rephrased the first sentence of the VQE
paragraph as follows: “We next employ the high-fidelity
gate set in the context of an actual application, in or-
der to provide a quantitative benchmark for future work
under realistic conditions.”
In terms of future prospects, we note that the accuracy
in the VQE experiment can be affected by errors in logic
operations, initialization and readout, and by the num-
ber of repetitions. In our experiment, readout errors were
the dominant contribution to the achieved accuracy of 20
mHa. The readout fidelity of the qubit states combined
was $\sim 80\%$. Although we have normalized the readout
window, we characterized the SPAM errors via GST and
attempted to minimize them (within the boundaries of
the setup used) about one day before the VQE experi-
ment. Slow voltage drift after calibration then typically
increases the SPAM errors. We point out that the read-
out of spin qubit, albeit not being a main focus in this
 work, need not be a bottleneck, as independent studies
 have shown readout fidelities of single spins above 99%
 (Zheng et al., Nat Nano 2019; Schaal et al., PRL 2020).
 We added a subsentence to clarify that the 20 mHa error
 is mainly attributed to drift in the readout parameters.
Nevertheless, the accuracy we have achieved is already
 comparable to similar work performed using trapped ion
 qubits with high-fidelity quantum logic (Hempel, et al.,
 PRX 8, 031022 (2018)). In addition, the performance
 of the simulation can be improved by utilizing a cus-
 tomized constant table (for $h_0 - h_5$) (O’Malley, et al.,
 PRX 6, 031007 (2016)). By comparison, we used the
 standard constant table generated by the OpenFermion-
 psi4 package, which leads to a conservative estimation of
 its performance.
In addition, the current manuscript does not seem to
 present a compelling validation of the custom fidelity
 analysis designed in this study. The authors used gate set
 tomography (GST) to characterize gate fidelities, which
 is established in the standard form; however, the one
 employed in this work is customized by the authors.
 Given that self-consistent tomographic reconstruction is
 a highly nonlinear and nontrivial estimation process,
 shouldn’t one naively expect the reliability to be influ-
 enced by choice of the gate set and model? Despite this,
 only a crude indicator of goodness of fit (four out of five
 stars) is given. The validation of their GST design and
 analysis would therefore deserve a more detailed discus-
 sion. Instead, a consistent measurement of a CPHASE
 gate fidelity by randomized benchmarking will help sup-
 port the claimed fidelity in a more accessible manner.
We notice that the first sentence in the method section
 “GATE SET TOMOGRAPHY ANALYSIS” reads “We
 designed a customized gate set tomography (GST)...”.
 We use the word “customized” only to emphasize that
 standard GST sequences can be generated using differ-
 ent native gate sets. This is different from a real cus-
 tomized GST experiment as explained below. As we are
 now aware of the confusion caused by this word, we re-
 moved it from the sentence.
The GST sequences used in our experiment is in fact
 generated in the standard form. There is no customiza-
 tion applied. Standard GST allows a fiducial pair re-
 duction (see Nielsen et al., arxiv:2009.07301) to remove
 redundant sequences. This redundancy originates from
 the fact that each germ only amplifies certain “direc-
 tions” in error space and the total number of directions
 is smaller than the corresponding space spanned by the
 combination of all fiducials. The software pyGSTi pre-
 computes all possible sequences and removes redundan-
 cies. This has been applied in our work and in Madzik et
 al. (arXiv:2106.03082), which was submitted to Nature
 at the same time as our manuscript. For clarity and com-
 pleteness, we point out that the GST procedure applied
 in Madzik et al. utilized a customized GST experiment
 with reduced fitting parameters, e.g. the assumed uni-
 tality of all processes automatically sets the first column

of all PTMs to be one and zero, which can be reasonable
 given the T_1 of a donor qubit is much longer. In contrast,
 the error model in our analysis contains all types of errors
 and the only constraint in our fitting is that each process
 must satisfy the CPTP condition.

The goodness of fit reported in the methods is only a
 pointer to the complete GST reports we have uploaded
 to Zenodo. The goodness of fit is computed by analyz-
 ing how many germ sequences deviate by more than N
 standard deviations from the expected outcome. In the
 complete report from the pyGSTi software, the model
 violation estimation of each sequence is reported along-
 side with the confidence region of each measured germ.
 To help the reader find this information, we have added
 “The total number of standard deviations exceeding the
 expected results for each L , as well as the contribution
 of each sequence to this number, can be found in the
 pyGSTi report, along with the supporting data.” to the
 methods section.

Regarding the choice between randomized benchmark-
 ing and GST, we note that in GST errors are considered
 as gate-dependent, which is the case not only for spin
 qubits but also for most other qubits. In contrast, ran-
 domized benchmarking (RB) assumes the errors to be
 gate-independent (the word “gate” here refers to Clifford
 gates instead of native gates). For small gate-dependent
 errors, RB is approximately still valid. However, for two-
 qubit Clifford gates, this assumption strongly breaks if
 the native single-qubit gates and two-qubit gates have
 errors of the same order of magnitude. In the present
 experiment, the errors of the native single-qubit gates
 are not much smaller than those of the CZ gate. There-
 fore, we argue that in this case a complete GST experi-
 ment is more powerful and reliable than RB. In fact, even
 for single-qubit RB, it has been reported multiple times
 that the single exponential decay model is broken in spin
 qubits (Fogarty et al. PRA 92, 022326 (2015); Kawakami
 et al. PNAS 113 (42) 11738-11743 (2016); Takeda et al.
 Nature Nano (2021)).

To conclude, given these issues above, I believe that
 this manuscript does not convincingly present the level
 of an advance that Nature readers will find as compelling
 as some of the other papers published by these same au-
 thors in this journal. I imagine that after these issues are
 addressed, the paper will be very well suited to a slightly
 more specialized journal.

We hope that the comments above have convinced the
 referee that the GST performed in this work is appro-
 priate and reliable for the fidelity analysis, and that the
 significance of the work meets the high standards of Na-
 ture.

Below I list some minor comments and suggestions.

(1) Does any observation (e.g. in Fig. 1(d)) indicate
 that avoiding a tilt is a good strategy for high-fidelity
 exchange-based gates in this device too, or is it assumed
 to be optimal based on previous studies?

Indeed, this is based on previous studies. It has been
 extensively investigated both experimentally (Martins et

al Phys. Rev. Lett. 116, 116801 (2016); Reed et al PRL 116, 110402 (2016)), and theoretically (Zhang et al PRL 118, 216802 (2017); Shim et al PRB 97, 155402 (2018)) that charge noise that couples to the exchange coupling can be minimized at the symmetry point.

(2) The single qubit gate fidelity in the full two-qubit space (99.16% on average) is reduced from the one in the single-qubit subspace. What is the reduction when only idling infidelities of 1% are considered? Do other mechanisms (such as crosstalk) contribute significantly at this stage?

We thank the referee for the insightful question. A good insight into which mechanisms are limiting here is given by the two-qubit error generator of the single-qubit gates from Extended Figure 1. If the only additional errors when going from the single-qubit space to the two-qubit space would be idling errors from uncorrelated dephasing of the idling qubit, the single-qubit gate fidelity would be reduced from 99.72% to 99.21%. This indicates that other mechanisms such as crosstalk do not significantly contribute to the drop in fidelity as compared to idling errors. To make this clear, we have added the sentence “By analysing the error generators, we find that errors from uncorrelated dephasing of the idling qubit dominate the drop in single-qubit gate fidelity when characterized in the two-qubit space. Coherent microwave-induced phase shifts, the main source of crosstalk errors, have been corrected by applying a compensating phase gate on the idling qubit (Extended Data Fig. 4)” into the paragraph discussing single-qubit fidelities.

(3) Can the authors consider reporting Bell state fidelities uncorrected for SPAM errors as well (in Fig. 4c and Extended Data Fig. 3)?

Accounting for SPAM errors, we estimate the Bell state fidelities to be 70.5% - 70.9%. This estimate uses the detailed SPAM matrices reported by the GST analysis, which can be found in the notebook uploaded together with the supporting data. We list the relevant information here. The initial state is analyzed to be (0.9776, 0.0168, 0.0047, 0.0009) (in the basis of $|00\rangle\langle 00|$, $|01\rangle\langle 01|$, $|10\rangle\langle 10|$, $|11\rangle\langle 11|$), and the matrix expressing the readout fidelity is

$$\begin{pmatrix} 0.8440 & 0.1141 & 0.1006 & 0.0151 \\ 0.0807 & 0.8096 & 0.0096 & 0.0881 \\ 0.0672 & 0.0115 & 0.7992 & 0.1237 \\ 0.0081 & 0.0647 & 0.0905 & 0.7731 \end{pmatrix}. \quad (40)$$

We emphasize that the main focus of this work was to demonstrate the high gate fidelities, and we didn't optimize the setup for high-fidelity readout. Therefore, we prefer not to report these numbers as they may give readers, especially non-experts, the wrong expression about the state of the art. Independent studies have demonstrated that high-fidelity readout of single spins is possible, as discussed also above.

(4) On line 72 of page 6 - what do the authors mean by the linear response regime?

Electric-dipole spin resonance (EDSR) is achieved by oscillating the electron in a magnetic field gradient. However, the oscillation amplitude of the electron position is not proportional to the driving amplitude, as the confinement potential is not perfectly harmonic. This results in a nonlinear dependence of the Rabi frequency on the driving amplitude. This effect has been reported before in the spin qubit community (Scarolino et al., PRL (2015), Scarolino et al., PRB (2017), Takeda et al., npj QInfo (2018)). The linear response regime here means that we apply a relatively small driving amplitude and therefore can ignore the distortion caused by nonlinear effects. To clarify this point, we have changed the sentence to “...in the linear response regime, where Rabi frequency is linearly dependent on the driving amplitude”.

(5) I believe there are typos in the domains of t given in Eq. (1).

We thank the referee for spotting the mistake in the domain of t . We corrected the typo.

(6) On line 36 of page 7, can the authors specify how many GST loops were (typically) required in this study?

We thank the referee for asking this. We have added the sentence “The presented results in **b** and **d** are achieved in four loops of correction, with each loop correcting the parameters by $\sim 70\%$ of the measured deviation.” in the caption of Extended Data Fig. 5

(7) The order of terms in Eq. (2) does not match the explanation given in the same paragraph. I also believe that \hbar in the definition of \vec{S} should be dropped.

Indeed the referee is correct. We fixed the mistake and swapped the order of terms in Eq. (2). We also dropped the factor \hbar in front of the spin matrices.

(8) On line 69 of page 7 - I suppose the word "amounts" should read "accounts".

We thank the referee for spotting the typo. We have corrected it.

(9) On line 78 of page 7 - I suppose the word "loose" should read "lose".

We thank the referee for spotting the typo. We have corrected it.

(10) On line 88 of page 7 - I suppose δf_{Q1} should be δf_{Q2} .

We thank the referee for pointing out the typo. We have corrected it.

(11) On line 10 of page 8, the authors obtained $\gamma \approx 1.2$. Is this fully consistent with the statement on line 4 of page 4?

We thank the referee for this insightful comment. Indeed the statement on line 4 of page 4 “shifts follow a linear relationship” is not entirely consistent with $\gamma \approx 1.2$. However, this value for γ is estimated by fitting the entire curve up to $J \sim 20\text{MHz}$, whereas for the CZ gate, the relevant regime extends only up to $J \sim 10\text{MHz}$, in which case $\gamma = 1$ is a good approximation, and this is used in simulations for simplicity. To make it clear, we have added the sentence “The micromagnet-induced single-qubit frequency shifts are approximated by linear functions within the voltage window of the CZ gate in the

1 numerical simulations.”

(12) There’s a typo in Eq. (12).
We thank the referee for spotting this important typo
and we have corrected it.
(13) In Eq. (18), information about the value of fre-
quency should be included, using a standard notation
used on line 33 of page 8.
We thank the referee, although we are not certain that
we understood the referee’s question correctly. We fol-
low here the approach outlined in Ref. [34] of the new
manuscript where the integral is identified by the (short
time-scale) Fourier transform of the input signal. This
utilizes insights from signal processing and spectral anal-
ysis. To avoid confusion with the noise power spectral
density that appears on line 33 of page 8, we changed the
notation in Eq. (18) to $S_s(f(t))$, representing the signal
energy spectral density. Furthermore, we replaced “re-
place” with “identify” in the manuscript below Eq. (18).
We also replaced the = symbol with the \propto symbol in Eq.
(18). We hope that these changes have addressed the
referee’s comment.
(14) Can the authors clarify the difference (in the
meaning) between a predicted gate infidelity of 0.022%
and an estimated fidelity $> 99.97\%$ on lines 29 and 37 on
page 9.
We thank the referee. Both numbers follow from
the same simulation and only differ by taking the lower
bound. We have omitted the second sentence, and moved
the first one to the place of the second sentence to make
our statement clear.
(15) On line 94 of page 9, there appears to be some-
thing wrong with the equation relating F_{ent} and F_{gate} .
We thank the referee. Indeed the formula is only valid
for the entanglement infidelity and the average gate infi-
delity. We corrected the formula.
(16) I encourage the authors to check Methods sec-
tion VIII again. Many parameters seem undefined (for
instance, p, q, r, s in Eq. (30) and σ , r_i and i in Eq. (31))
and I believe there are some typos (at least in Eqs. (33),
(34) and (35) and possibly in Eq. (31)). Furthermore, I
am confused from which side the qubit index is counted
after the authors reduced the dimension using the BK
transformation and implicitly relabeled the qubits. (It
should be consistent with the main text, e.g. Fig. 5b for
readability.)
We thank the referee for the comment. We have
checked and corrected the typos and have added addi-
tional information for clarification.
(17) Extended Data Fig. 3 - there’s a typo in the Bell
state in the figure caption for panel (c). Can the authors
briefly explain why the estimated state fidelities are the
same for Ψ^+ and Ψ^- (Φ^+ and Φ^-) - i.e. any reasons why
the estimated fidelity of virtual Z gate is 1?
We thank the referee for this insightful comment. We
have fixed the typo. As the name suggest our virtual Z
gate is only a phase update on the microwave reference
clock of the qubit which consumes no time and does not
introduce errors. We clarified this in the figure caption
by changing the sentence to “ $Z_{Q_i}^2$ is a virtual π -rotation
around the \hat{z} axis on the i th qubit, which is executed by
a phase update on the microwave reference clock of the
qubit and therefore is error-free.”
(18) Extended Data Fig. 5 - can the authors give error
bars in panels c and d and explain (preferably in the main
text) how they derived the error bars for the reported
value of CPHASE fidelity?
We thank the referee. We have added the error bars
into the panels and have added the sentence “Error bars
displayed here and elsewhere show the $2\sigma \approx 95\%$ confi-
dence intervals computed using the Hessian of the log-
likelihood function.” into the main text (page 4). We
also add a similar statement in the caption of Extended
Data Fig. 5.
Referee nr. 2 (Remarks to the Author):
In the manuscript "Computing with spin qubits at the
surface code error threshold," the authors report a spin-
based two-qubit quantum processor with average single-
and two-qubit gate fidelities above 99%, reaching for the
first time the generic error threshold for the surface code
in this system. Using standard Ramsey interference ex-
periments and gate set tomography, the authors provide
detailed characterizations of device properties as well as
the full Pauli transfer matrices and error generators de-
scribing the primitive quantum gate set used. They then
go on to demonstrate the capabilities of their device by
running a variational quantum eigensolver algorithm and
computing the dissociation energy for molecular hydro-
gen.
The achievement of a spin-based quantum processor
with two-qubit gate fidelities above 99% represents a
major milestone for solid-state semiconductor quantum
computing platforms. Additionally, the manuscript is
detailed, clearly written, and well-organized. I also find
the detailed characterization of the device and the vari-
ous noise sources helpful for the community to perform
follow-up theoretical work on this platform. For these
reasons, it meets the high standards of Nature provided
the following points are addressed:
We thank the referee for reviewing our manuscript and
for the supportive comments. We give our response to the
questions below.
1. Some aspects may be hard to follow for the spin
qubit community. While verification and validation are
critical aspects of building a quantum processor, not ev-
eryone is familiar with some of the relevant concepts.
More specifically, the readers would benefit by under-
standing better the concept of germ sequences and how
these are chosen.
We agree with the referee that GST is a complex proto-
col. As it is a well validated approach and has been exten-
sively investigated, we focus mainly on the aspects of the
design and the results of the GST instead of reintroduc-
ing it. To make the text more accessible, we made a few
changes. We have rephrased a sentence in the paragraph
describing GST in page 2 as "Germs are short sequences
of gates taken from the universal gate set (see Methods).
They are repetitively executed to amplify different types
of gate errors in the gate set, such that SPAM errors
can be isolated". We have also added a more detailed
description about germ selection in the methods section.
We also made some changes in the method section about
VQE to make it more clear.
2. The text refers to Figure 1e, however, there is not
(e) panel in Fig. 1.
We thank the referee and we have put in the correct
label 1d.
3. I find the last sentence of the abstract a bit odd and
would suggest to the authors to rephrase it. In particular,
the comment that now spin qubits 'have gained credibil-
ity' is not a scientific statement, and it's something that
the authors cannot claim.
We thanks the referee for the critical comment. We
accept the referee's suggestion and have changed the sen-
tence to "Having surpassed the 99% barrier for the two-
qubit gate fidelity, semiconductor qubits are well posi-
tioned on the path to fault-tolerance and to possible ap-
plications in the era of noisy intermediate-scale quantum
(NISQ) devices."
4. It is interesting and helpful that the authors charac-
terize single-qubit gate fidelities in the two-qubit system.
As they say, this fidelity is distinct (lower) from single-
qubit gate fidelities in a single-qubit system. For the
same reason, the two-qubit gate fidelity will be lowered
in a three-qubit (or larger) system. It would be help-
ful to have a relevant comment about this point and its
repercussions for scalability. Given what the present ex-
periment has shown, can the authors speculate about the
expected two-qubit fidelity in a larger system?
We thank the referee for this excellent comment. In-
deed one can expect the fidelity of the two-qubit gate to
be reduced if measured in a larger Hilbert space. Fidel-
ities generally are reduced when including a larger state
space due to both crosstalk errors and dephasing errors
on the idling qubits. We note that the CZ gate that we
have implemented here is only based on local gate volt-
age pulses, with individually tunable tunnel couplings,
hence spin exchange couplings. Utilizing the "virtual
gate" technique, crosstalk on the quantum dot electro-
chemical potentials (this work) and on other tunnel cou-
plings in a larger array can be well compensated, such
that there are minimal crosstalk errors. As for the de-
phasing errors on idling qubits, if some qubits need to
be idle for a relatively long period, decoupling pulses can
be applied to reduce the errors. To highlight this mes-
sage, we have added "The virtual barrier gate technique
used here efficiently suppresses crosstalk errors during
two-qubit gates. Therefore, we expect the CZ fidelity to
be mostly affected by dephasing errors of idling qubits in
a larger space which can be corrected for with decoupling
pulses" into the paragraph discussing the CZ fidelity.
5. Typically, the CPHASE gate with the phase = π is
called CZ. It is probably clearer to use this more standard
term.
We agree with the referee and now refer to the gate as
"CZ".
6. The theoretical Bell state fidelity is reported, but
the experimental number is not given. Why is that?
We didn't perform experimental state tomography of
the Bell states because the focus of this work is on the
gate process instead of the states, although we could have
done it. Still we found it useful and interesting to the-
oretically estimate the Bell states. Importantly, the de-
tailed and complete tomographic information extracted
from GST makes it possible to emulate the performance
of any quantum circuit. This estimation also provides a
good example of a possible application of GST.
7. While overlapping gates in the QD device have been
shown by other groups to lower the noise, the authors
don't seem to have adopted it. Is there a reason for that?
There is no specific reason to use a single-layer gate
design rather than overlapping gates. This design was
chosen for historical reasons related to the development
of device fabrication in our group. At this moment we
are also able to make devices with overlapping gates. As
for the noise screening effect, in the device presented,
there is a dedicated screening gate which is placed on
top of all the gates shown in the SEM, but underneath
the micromagnet. This screening gate covers the entire
double dot area and contributes to screening of charge
noise. More details have been published in the methods
section in Xue et al. (Nature 593, 205–210 (2021)), where
the same device was used. We have added additional
information “Scanning electron microscope images of a
similar device to that used here showing the quantum
dot gate pattern and the micromagnet on top (the device
used in the experiment has an additional screening gate
above the fine gates [17])” into the caption of Figure 1.
8. In the Methods section, it is mentioned that the
Ramsey sequences “are automatically executed every two
21 hours”. This statement is unclear to me—this seems like
a long timescale, where does it come from?
We thank the referee for pointing this out. Given
that the qubit frequencies actually remain very stable,
we only need to perform a frequency calibration at the
beginning of each GST experiment (every 4 hours). We
insert another Ramsey calibration in the middle to dou-
ble check the frequencies, and typically find shifts of only
a few tens of kHz. Importantly, between frequency ad-
justments, we average over all 1685 sequences to ensure
that this procedure does not effectively introduce non-
Markovian noise. We have added “at the beginning and
the middle (after 100 times average of each sequence) of
the GST experiment” in the corresponding sentence.
9. Typo in Sec. III: “loose information” should be
“lose information”.
We thank the referee for spotting the typo. We have
corrected it.
10. In Sec. VII some gate durations are given, but
others are not.
We thank the referee for pointing out the missing in-
formation. We have added “with durations of 150 ns and
200 ns respectively” in to the beginning of Sec. VII.
11. When UCC is introduced, the authors write that
it “is widely believed to be a powerful approach”. I would
say that UCC is a standard approach used in experimen-
tal proof of principle demonstrations, but it is far from
state-of-the-art in the theoretical VQE community.
We thank the referee for the helpful comment. We
removed this phrase and changed the sentence to “We
apply the unitary coupled cluster (UCC) theory to the
parameterized ansatz state, which is used to describe
many-body systems and cannot be efficiently executed
on a classical computer [53].”
Referee nr. 3 (Remarks to the Author):
In this manuscript the authors experimentally probe
the performance of a 2 qubit quantum information pro-
cessor implemented in a silicon quantum dot double well.
The qubits are defined as single electron spin states,
one per well. Single qubit gates are performed through
EDSR and the two qubit interaction comes from electro-
statically modulating the tunnel barrier in order to in-
duce an exchange interaction between the two spins. The
authors use gate set tomography to measure two qubit
gate fidelities greater than 99% and proceed to run a
small variational quantum algorithm.
There have been a couple demonstrations of 2-qubit
gates in semiconductor systems, but as far as this referee
is aware this is currently the highest fidelity in dots by
a fair margin and probably the first demonstration that
puts semiconductor qubits in contention with supercon-
ductor or ion quantum computing implementations. For
that reason I recommend this for publication in Nature
with a couple minor revisions.
We thank the referee for reviewing our manuscript and
for the positive recommendation. We respond to the
other comments of the referee below.
I suppose I have some bias that spin qubits are broadly
interesting, and that an experiment at the forefront of
spin qubit implementations should be a shoe-in for Na-
ture. With that perspective, my general comments to
the authors to improve this manuscript would be an ex-
panded discussion about what challenges remain in order
to make this a truly competitive quantum computation
platform.
1. Gasetomography is a fine method for extract-
ing gate error but I find it odd that the authors haven't
shown any results from randomized benchmarking (RB).
For some time now RB has basically been the industry
standard for reporting the performance of one and two
qubit devices and omitting any results seems like an over-
sight.
I find this omission troubling because I can imagine
several reasons why it might be hard to run RB on a
dot system. DC electronics can be tricky, and in order to
quantify errors at the $1e-3$ level (which probably amounts
to 1000's of Clifford gate) I could imagine issues with sta-
bility or waveform memory. If long sequences of gates are
hard for some reason I still absolutely believe this result
should be in Nature, but I think it would behoove the
authors to point out the open research and engineering
problems in this promising platform. If there was no rea-
son not to do RB in the first place, then do it, and the
results will be all the stronger.
We thank the referee for the insightful comment. We
chose to implement GST instead of RB for a few reasons:
First, so far standard and interleaved RB have been
widely used for its simple analysis process. The fidelity
can be extracted by fitting a single exponential decay
curve. However, this model of depolarizing noise can be
broken when there are large gate-dependent errors. It is
often omitted that the word "gate" here refers to Clifford
gates instead of native gates. For single-qubit Clifford
gates, this model violation is small due the simple com-
pilation. However, for two-qubit Clifford gates, especially
when using a CZ gate as the native one, the complexity in
the compilation can lead to large gate-dependent errors.
In some published work from the superconducting qubit
and trapped ion qubit community, the native single-qubit
gates are about 1-2 orders of magnitude better than the
two-qubit gates used, which reduces the effect of gate-
dependent errors. However, this is not the case in our
work because the single-qubit gates are not much bet-
ter than the two-qubit gate. In contrast, GST does not
make such an assumption. In fact, GST assumes that
all native gates can have different errors. Therefore, we
believe that GST can provide a more accurate analysis
of the fidelities.
Second, as the referees have pointed out, to extract
such high fidelity numbers, RB sequences compiled by a
large number of two-qubit Clifford gates are needed. Re-
garding potential engineering problems, our control hard-
ware can fully accommodate the long sequences needed
for two-qubit RB. However, an issue with applying long
sequences is slow drift of the qubit parameters, such as
frequency shifts induced by microwave heating or uncom-
pensated bias-tee charging effects. At this moment, heat-
ing effects are still a dominating error source in the read-
out as heating directly affects the electron reservoir and
the charge sensor (these issues are not fundamental and
can be mitigated in principle, for instance by gate-based
Pauli spin blockade readout). The shift of the readout
visibility and the qubit frequencies is often a function
of the total duration of microwave burst, and therefore
behaves as non-Markovian noise in an RB experiment.
It has been extensively investigated that non-Markovian
noise in spin qubit systems can break the depolarizing
noise model even for single-qubit RB and thus cannot be
fitted by a single exponential decay curve (Fogarty et al.,
PRA 92, 022326 (2015); Kawakami et al., PNAS (2016);
Takeda et al., Nature Nano (2021)). An example that we
have seen in our experiment is that changes in the read-
out visibility with increased length of microwave bursts
as needed for RB experiments can affect the decay.
To make these points clear, we have added "In ad-
dition, GST accounts for gate-dependent errors." in the
main text, and "We note that the sequences used in GST
are shorter than the sequences involved in other methods
to self-consistently estimate the gate performance, such
as randomized benchmarking. As a result, GST suffers
less from drift in qubit frequencies and readout windows
induced by long sequences of microwave bursts" into the
methods section of gate set tomography analysis.
2. When I first saw the title of this work I thought I'd
been handed a paper out of a time-capsule from the su-
perconductor community, where there were manuscripts
like "Superconducting quantum circuits at the surface
code threshold for fault tolerance" out of the UCSB (later
Google) group and "Universal Quantum Gate Set Ap-
proaching Fault-Tolerant Thresholds with Superconduct-
ing Qubits“ from IBM. If this was intended to be a cute
reference to those works it’s fine, but I think there are
some lessons to be drawn from them as well.
So, short of these titles being a touch hyperbolic, the
main issue is that neither IBM nor Google have demon-
strated fault-tolerance despite being at the “threshold”
7-8 years ago. The issue, of course, has been scaling.
There was a promise in superconducting qubits, similar
to those made by some semiconductor groups, that scal-
ing is somehow a straightforward engineering problem
once one has demonstrated high fidelity one and two-
qubit gates. It is a little hard to look at the pictures of
the device in fig 1 and immediately see how this technol-
ogy is scalable. Perhaps some discussion of the challenges
moving forward in terms of engineering and fabrication
would be interesting to the broader audience.
We thank the referee for the comment and the ref-
erree is correct in our homage to the “old” papers. We
are very aware that neither Google nor IBM or any
other group until now have shown fault-tolerance despite
having achieved gate fidelities above the fault-tolerance
threshold. In this work, as well as in the works that the
referee mentioned, the main message is that at the fun-
damental physics level, there is no limitation to achieve
high-fidelity quantum logic, which is crucial for future
fault-tolerant quantum computing. As suggested by the
referee, one of the biggest challenges in achieving fault-
tolerance is scaling, which depends on the architecture,
the connectivity, the uniformity, and the individual phys-
ical error rates. To specifically point out the challenges
for further studies, we have added “Combining high-
fidelity initialization, readout, and control into a demon-
stration of fault-tolerance poses several key challenges to
be overcome. First, sufficiently large and reliable quan-
tum dot arrays must be constructed, with good connec-
tivity between the qubits. Second, the fidelities achieved
in small-scale systems must be maintained across such
larger systems, which will require reducing idling and
crosstalk errors” into the conclusion paragraph.

Reviewer Reports on the First Revision:

Referee #1 (Remarks to the Author):

I appreciate the feedback and changes the authors have made to address the issues raised by the reviewers. I am supportive of publishing this manuscript in Nature more or less as is except the title.

In the previous round of review, I pointed out that in addition to errors in single- and two-qubit gate control, errors in and time for initialization and readout need to be taken into account together, in order to judge if the computation as a whole would be below the fault-tolerance threshold. In their response, the authors agreed by saying "our main claim is that the error rates of the native gate set achieve the most widely used threshold value required to achieve fault-tolerance and that there is no limitation at the fundamental physics level for spin qubits to reach such high gate fidelities. This is clearly different from claiming a system to be fault tolerant." I appreciate that they acknowledge the difference and made changes to make their claim explicit. However, the title still implies that the claim is (quantum) computing with spin qubits at the surface code error threshold (for fault tolerance). This is misleading and needs to be reworded to be consistent with their focus on control fidelity.

I have following comments for improvements that I would like the authors to consider (but I won't hold publication contingent on these points):

1) Related to my previous comment (3): I understand why the authors prefer not to report the Bell fidelities after accounting for SPAM errors, especially to non-experts, but if so, it is better to state clearly, for non-expert readers, that the Bell fidelities they report do not include SPAM errors (on line 57 of page 4).

2) Related to my previous comment (13) on Eq. (18): I would like to explain what I meant in more detail. $S_{\{\Delta v_B\}}(\omega)$ in the previous section denotes the energy spectral density, ESD, of Δv_B at the angular frequency of ω (to be honest, $S_{\{v_B\}}(\omega)$ would make more sense to me here, as we are interested in the noise in v_B). This follows a standard notation. Now, in the same notation, $S_S(f(t))$ in Eq. (18) would mean the ESD of S at the angular frequency of $f(t)$. This is, however, not the case. If I am not mistaken, the expression here should be something like $S_f(t) (\Delta E_z/\hbar)$, where $\Delta E_z/\hbar$ is the value of angular frequency at which ESD is measured.

3) I appreciate that the authors have checked carefully and improved Methods section VIII. I have some further comments for consideration:

3a) Do you really need an * on ψ_q in Eq. (31)?

3b) I am confused by the qubit index used in this section. Does the Z gate in $h1Z1$ and $h1ZI$ terms in Eq. (35) really act on the same qubit? (Shouldn't the $h1ZI$ term actually correspond to $h1Z2$ instead?) Put differently, after the HF initial state 0001 is reduced to 01, what is the index of the qubit initialized in state 1? After ignoring qubits 2 and 4 (both are always in the 0 state) and relabeling qubit 3 as qubit 2, the answer is 1. It is however 2 in Fig. 5 (and everywhere else).

4) I believe I found typos in the units of the parameters β_1 and β_2 in Methods section IV (line 33 of page 8).

Referee #2 (Remarks to the Author):

The authors have addressed the questions and concerns of the three referees. I find that the manuscript has improved considerably following the revisions, and I can now recommend publication in Nature.

Referee #3 (Remarks to the Author):

I'm fine with the changes the authors have made and think this manuscript is suitable for publication in Nature as is.

As a side note, since I guess I can't help myself, surely the Markovianity assumptions that go into deriving a FT threshold are invalidated by the drifts in "qubit frequencies and readout windows" in this system that make RB challenging and so describing this work as "at the surface code error threshold" is a bit disingenuous.

Referee nr. 1 (Remarks to the Author):
I appreciate the feedback and changes the authors have
made to address the issues raised by the reviewers. I am
supportive of publishing this manuscript in Nature more
or less as is except the title.
We thank the referee for reviewing the manuscript
and for supporting publication of our manuscript in Na-
ture. We give our response below and have modified our
manuscript accordingly.
In the previous round of review, I pointed out that
in addition to errors in single- and two-qubit gate con-
trol, errors in and time for initialization and readout need
to be taken into account together, in order to judge if
the computation as a whole would be below the fault-
tolerance threshold. In their response, the authors agreed
by saying "our main claim is that the error rates of the
native gate set achieve the most widely used threshold
value required to achieve fault-tolerance and that there
is no limitation at the fundamental physics level for spin
qubits to reach such high gate fidelities. This is clearly
different from claiming a system to be fault tolerant."
I appreciate that they acknowledge the difference and
made changes to make their claim explicit. However, the
title still implies that the claim is (quantum) comput-
ing with spin qubits at the surface code error threshold
(for fault tolerance). This is misleading and needs to
be reworded to be consistent with their focus on control
fidelity.
We see the referee's point that the term "computing"
may be understood more broadly than referring only to
the sequence of quantum logical operations. The lat-
ter is what we intended. We therefore changed the title
to "Quantum logic with spin qubits crossing the surface
code threshold".
I have following comments for improvements that I
would like the authors to consider (but I won't hold pub-
lication contingent on these points):
1) Related to my previous comment (3): I under-
stand why the authors prefer not to report the Bell fi-
delities after accounting for SPAM errors, especially to
non-experts, but if so, it is better to state clearly, for
non-expert readers, that the Bell fidelities they report do
not include SPAM errors (on line 57 of page 4).
We agree with the referee and have added "neglecting
SPAM errors" in the sentence.
2) Related to my previous comment (13) on Eq. (18):
I would like to explain what I meant in more detail. $S_{\delta v_B}$
(ω) in the previous section denotes the energy spectral
density, ESD, of δv_B at the angular frequency of ω (to
be honest, $S_{v_B}(\omega)$ would make more sense to me here,
as we are interested in the noise in v_B). This follows a
standard notation. Now, in the same notation, $S_S(f(t))$
in Eq. (18) would mean the ESD of S at the angular
frequency of $f(t)$. This is, however, not the case. If I am
not mistaken, the expression here should be something
like $S_f(t) (\Delta E z / \hbar)$, where $\Delta E z / \hbar$ is the value of angular
frequency at which ESD is measured.
We thank the referee for elaborating their question and
spotting this typo. We have corrected this. The referee is
right that following the standard notation for a standard
Fourier transform one would arrive at $S_f(t) (\Delta E z / \hbar)$.
However, in our experiment $\Delta E z$ is a fixed parameter
and what is varied is the pulse length t_p (boundaries of
the short time-scale Fourier transform window) or the
shape of the pulse J . We therefore keep the notation to
emphasize the parameters that are varied.
3) I appreciate that the authors have checked carefully
and improved Methods section VIII. I have some further
comments for consideration:
3a) Do you really need an * on ψ_q in Eq. (31)?
The referee is correct about this typo. We have now
corrected it.
3b) I am confused by the qubit index used in this sec-
tion. Does the Z gate in h1Z1 and h1ZI terms in Eq. (35)
really act on the same qubit? (Shouldn't the h1ZI term
actually correspond to h1Z2 instead?) Put differently,
after the HF initial state 0001 is reduced to 01, what is
the index of the qubit initialized in state 1? After ig-
noring qubits 2 and 4 (both are always in the 0 state)
and relabeling qubit 3 as qubit 2, the answer is 1. It is
however 2 in Fig. 5 (and everywhere else).
We thank the referee for pointing out this important is-
sue. We notice in that particular paragraph we wrote the
encoded qubit state in the form of $|Q_4 Q_3 Q_2 Q_1\rangle$, whereas
we wrote it in the form of $|Q_1 Q_2\rangle$ elsewhere. Now we
have corrected it and have clarified it by adding the sen-
tence "qubit 1 has been relabeled as qubit 2 and qubit
3 has been relabeled as qubit 1. The HF initial state is
therefore reduced to $|01\rangle$ ".
4) I believe I found typos in the units of the parameters
beta1 and beta2 in Methods section IV (line 33 of page
8).
We thank the referee for spotting these typos. We have
corrected it in the revised version.
Referee nr. 2 (Remarks to the Author):
The authors have addressed the questions and concerns
of the three referees. I find that the manuscript has im-
proved considerably following the revisions, and I can
now recommend publication in Nature.
We thank the referee for reviewing our manuscript and
for recommending publication in Nature.
Referee nr. 3 (Remarks to the Author):
I'm fine with the changes the authors have made and
think this manuscript is suitable for publication in Nature
as is.
As a side note, since I guess I can't help myself, surely
the Markovianity assumptions that go into deriving a FT
threshold are invalidated by the drifts in "qubit frequen-
cies and readout windows" in this system that make RB
challenging and so describing this work as "at the surface
code error threshold" is a bit disingenuous.
We thank the referee for reviewing our manuscript
and for recommending publication in Nature. As in
the response to previous round of referee comments, we
acknowledge that any error correction threshold comes
with many assumptions, for instance on the connectiv-
ity, on the fidelities not worsening as the number of
qubits is increased, and indeed also on the nature of
the noise (Markovianity, biased errors, spatial correla-
tions, etc). This is also captured by the statement
“under certain assumptions” in the introductory para-
graph of the manuscript. If some assumptions are re-
laxed or not met, this may decrease or increase the fault-
tolerance threshold, depending on the situation. System-
atic low-frequency drifts can be easily accounted for in
the calibrations, for instance. The 99% number is thus
merely to be understood as a widely used benchmark,
and much follow-up work will be needed to demonstrate
fault-tolerance, as indicated explicitly in the concluding
paragraph of the manuscript.